# Mid-late Holocene vegetation history of the Argive Plain (Peloponnese, Greece) as inferred from a pollen record from ancient Lake Lerna

Cristiano Vignola[1,2]*, Martina Hättestrand[3], Anton Bonnier[4], Martin Finné[4,5], Adam Izdebski[1,6], Christos Katrantsiotis[3,7], Katerina Kouli[1,8], Georgios C. Liakopoulos[1], Elin Norström[3,9], Maria Papadaki[1], Nichola A. Strandberg[3,10], Erika Weiberg[4], Alessia Masi[1,2]

1 Palaeo-Science and History (PS&H) Independent Research Group, Max Planck Institute for the Science of Human History, Jena, Germany, 2 Department of Environmental Biology, Sapienza University of Rome, Rome, Italy, 3 Department of Physical Geography, Stockholm University, Stockholm, Sweden, 4 Department of Archaeology and Ancient History, Uppsala University, Uppsala, Sweden, 5 Department of Social and Economic Geography, Uppsala University, Uppsala, Sweden, 6 Institute of History, Jagiellonian University in Krakow, Krakow, Poland, 7 Environmental Archaeology Laboratory, Department of Historical, Philosophical and Religious Studies, Umeå University, Umeå, Sweden, 8 Department of Geology and Geoenvironment, National and Kapodistrian University of Athens, Athens, Greece, 9 Department of Geological Sciences, Stockholm University, Stockholm, Sweden, 10 School of Geography and Environmental Science, University of Southampton, Southampton, United Kingdom

* vignola@shh.mpg.de

**Data Availability Statement:** Pollen data are available from the Neotoma database (DOI: https://doi.org/10.21233/6A4G-5105).

## Abstract

This study provides a high-resolution reconstruction of the vegetation of the Argive Plain (Peloponnese, Greece) covering 5000 years from the Early Bronze Age onwards. The well dated pollen record from ancient Lake Lerna has been interpreted in the light of archaeological and historical sources, climatic data from the same core and other regional proxies. Our results demonstrate a significant degree of human impact on the environments of the Argive Plain throughout the study period. During the Early Bronze Age evidence of a thermophilous vegetation is seen in the pollen record, representing the mixed deciduous oak woodland of the Peloponnesian uplands. The plain was mainly used for the cultivation of cereals, whereas local fen conditions prevailed at the coring site. Towards the end of this period an increasing water table is recorded and the fen turns into a lake, despite more arid conditions. In the Late Bronze Age, the presence of important palatial centres modified the landscape resulting in decrease of mixed deciduous oak woodland and increase in open land, partly used for grazing. Possibly, the human management produced a permanent hydrological change at Lake Lerna. From the Archaic period onwards the increasing human pressure in association with local drier conditions caused landscape instability, as attested by a dramatic alluvial event recorded in the *Pinus* curve at the end of the Hellenistic Age. Wet conditions coincided with Roman times and favoured a forest regeneration pattern in the area, at the same time as we see the most intensive olive cultivation in the pollen record. The establishment of an economic landscape primarily based on pastures is recorded in the Byzantine period and continues until modern times. Overgrazing and fires in combination with arid

**Funding:** The initiation of this project was financed by the Swedish Research Council (grant number 621-2012-4344; P.I. Karin Holmgren; https://www.vr.se/english.html) through which MH, CK and EN have been supported. Pollen analysis carried out by CV and NAS, as well as further dating, have been financed by the Bolin Center for Climate Research, Stockholm University (https://bolin.su.se), the PS&H Group of the Max Planck Institute for the Science of Human History (https://www.shh.mpg.de/1056512/psh), and The Royal Swedish Academy of Letters, History and Antiquities (Enboms donationsfond; https://www.vitterhetsakademien.se/english/the-royal-swedish-academy-of-letters-history-and-antiquities.html). EW, MF and AB have been supported by the Swedish Research Council (grant nos. 421-2014-1181 and 2019-02868). All the funders had no role in study design, data collection and analysis, decision to publish, or preparation of the manuscript.

**Competing interests:** The authors have declared that no competing interests exist.

conditions likely caused degradation of the vegetation into garrigue, as seen in the area of the Argive Plain today.

## Introduction

The Peloponnese (Fig 1A) is a key region for the study of the long-term interaction between human societies and the environment [1–3]. The landscapes of the peninsula have been formed through tectonic activities and fluvial erosion and deposition of sediments, in combination with sea level variations and climate changes [4–6]. Contemporary vegetation is defined by high variability due to a large variation in parameters such as geomorphology, soil type and hydrology, in combination with gradients of temperature and precipitation across the peninsula. Moreover, the Peloponnese, and southern Greece in general, has been strongly affected by human activities since at least the Bronze Age, which contributed to create cultural regions surpassing environmental differences [2, 3].

In this paper the role of palynology in investigating natural and human-forced changes of past environments is stressed. Although pollen analysis provides relatively local vegetation histories, these histories often record human-environmental dynamics on regional and super-regional scales, as broadly shown in the Mediterranean [7–11]. We focus on the mid-late Holocene vegetation of the Argive Plain in the northeastern Peloponnese (from 4760 BP/2810 BCE to present). The catchment area of the cored sediment basin—hence our study area—covers the lowlands of the Argive Plain proper, as well as the surrounding foothills and uplands (Fig 1B). The rich evidence for intense human occupation, which altered the landscape from the Neolithic to the present, has spurred several studies on local palaeoenvironmental changes. In particular, these previous studies have focused on the role of geomorphological changes and vegetation dynamics in relation to socio-economic development throughout the Holocene [12–19]. However, significant methodological advancement within paleoenvironmental sciences during the last two decades, provide expanded possibilities for a detailed study of the relationship between palaeoclimatic fluctuations and the evolution of the local plant ecosystem. In this study, our specific aims are (i) to present a high-resolution record of past vegetation change in the area of the Argive Plain, and (ii) compare pollen data with complementary archaeological, historical, climatic, and palaeoenvironmental proxies at a local and regional scale.

## Study area

### Geomorphology and hydrology of the Argive Plain

The Argive Plain (Fig 1B) is located at the carbonate basin of the Gulf of Argos and is surrounded by hills formed by karstified limestones with steep slopes rising to 400–700 m [18]. The main streams are the Inachos River, flowing in the central plain and drying out during summer, and the Erasinos River to the west, which is perennial.

The plain, surrounded by highland massifs to the east and west, can be divided into different geomorphological zones: a horseshoe-shaped lowland extends to the north from 25 to 100 m a.s.l., consisting of Pleistocene alluvial fans; in the coastal portion from Argos to Nafplio (E) and Lerna (W) the gradation to the sea is extremely low (0–25 m a.s.l.) and the sediment deposits are mostly represented by Holocene alluvia [18]. Most of them are streamflood deposits and overbank loams derived from the running waters originating in the highlands which cause extensive but infrequent flooding events.

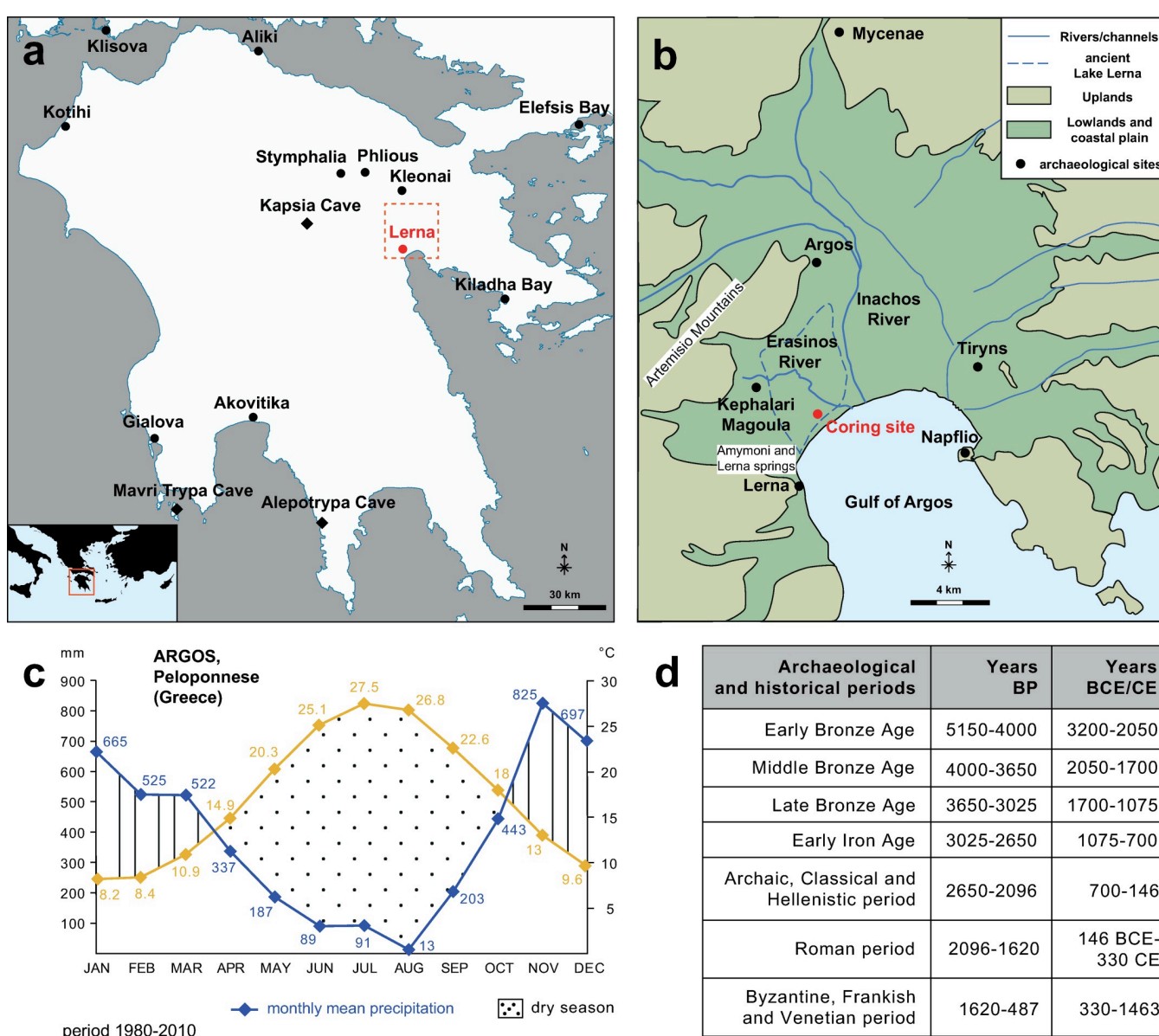

**Fig 1. Study area in the Peloponnese (Greece) highlighting the Argive Plain with ancient Lake Lerna and its climatic data, as well as the location of sites for the investigated periods.** (a) Location of Lake Lerna and other Peloponnesian sites mentioned in the paper. Dots stand for palaeoenvironmental records, diamonds for palaeoclimatic ones.; (b) Detail of the Argive Plain with the former lake and coring site. The main archaeological and historical sites are shown; (c) Monthly mean temperature (˚C) and precipitation (mm) values recorded in Argos during the period 1980–2010 [32]; (d) Chronology of the main archaeological and historical periods in Greece (from [2]).

The geomorphology of the area has thus been shaped mainly by fluvial processes, along with the transgression and regression of the sea due to coastal accretion during the Holocene. The maximum transgression of the sea, i.e. by the Gulf of Argos, has been dated to before 5500 BP, at which time a portion of the eastern coastal plain was submerged [18, 19]. The southwestern portion was instead occupied by Lake Lerna (Fig 1B): a freshwater basin permanently supplied by the Erasinos River and separated from the sea by a beach barrier formed at ca. 8300 BP [17, 18]. At ca. 6500 BP the lake reached modern Argos and covered an area of 4.2

km$^2$ with a water depth of about 7 m [12, 20]. Between ca. 5000–4000 BP the eastern part of the lake basin was filled because of accretion of sediments carried by the Inachos River, whereas the western part survived as a shallow water body [17]. Apart from the Erasinos River, the water regime of the lake was mainly determined by the coastal springs of Lerna and Amymoni (Fig 1B), as well as by some low-discharging streams from the eastern slopes of the Artemisio Mountains. A small wetland, flooded during the rainy seasons, is the only present trace of the former Lake Lerna.

The coring site chosen for this study is located in the southernmost part of the area of the ancient lake (Fig 1B). Sedimentation rate and pollen deposition can vary to a high degree in these kinds of shallow wet environments close to rivers, where the water table and sediment inflow commonly fluctuate (e.g. [21–23]). During phases of local fen conditions marshy taxa can be over-represented in pollen records while lake conditions generally better reflect the vegetation in a larger area. In a relatively open environment, such as the Argive Plain, we would expect the pollen record from the coring site to mainly reflect the vegetation on the Argive Plain including the slopes of the surrounding mountains.

## Vegetation

The vegetation of the region includes a variety of plant communities as generally described for Greece [24, 25]. In the highlands, mountain broadleaved and coniferous forests prevail; *Abies cephalonica Loudon*, endemic in the central and southern Peloponnese, is mainly present in association with *Pinus nigra J. F. Arnold* ([26]; see also the modern pollen data provided by [13] and [27]). In the upland valleys and hill slopes the thermophilous mixed deciduous forest dominates. The alliance of deciduous *Quercus L.*, *Ostrya carpinifolia* Scop. and *Carpinus orientalis* Mill. prevails, but Mediterranean species also occur (*Pistacia terebinthus L.*, *Phillyrea latifolia L.*, *Prunus mahaleb L.*). The lowlands are covered by the Mediterranean sclerophyllous vegetation which belongs to the evergreen forest of oaks (*Quercus coccifera L.*, *Q. ilex L.*) and by pines (*Pinus brutia Ten.*, *P. halepensis Mill.*, *P. pinea L.*). Since the Argive Plain has been strongly shaped by human activities, one of the main degraded environments is the phrygana, which is a semi-shrub heathland of garrigue type [28]. More open natural communities are characterised by *Olea europaea L.*, *Pistacia lentiscus* L. and *Ceratonia siliqua* L. in association with sclerophyllous shrubs (*Phillyrea latifolia L.*, *Myrtus communis L.*, *Rhamnus alaternus L.*, *Pistacia terebinthus L.*) [25]. The hygrophilous vegetation of the wetlands mainly consist of *Phragmites australis (Cav.) Steud.* and *Arundo donax L.*, whereas trees only occur in few isolated patches (i.e. *Eucalyptus L'Hér*, *Salix L.*, *Populus L.*, *Cupressus L.*). Perennial (e.g. *Typha latifolia L.*) and floating (*Cladium mariscus (L.) Pohl.*, *Nymphaea L.*) macrophytes grow in ponds and are replaced by halophytes at the Inachos River mouth, where *Tamarix L.* is widespread [28].

Two pollen studies have previously been published from Lake Lerna. The most comprehensive one is that of Jahns [13], presenting a continuous pollen record of the last 5000 years. The author describes a vegetation history that begins with a mixed deciduous oak woodland growing in the uplands and an open Mediterranean forest consisting of *Quercus ilex L./Q. coccifera* L. dominating the plain, where there were several pre-existing human settlements. Thereafter, the deciduous forest became denser at lower altitudes, replacing the evergreen vegetation. This is followed by a decrease of the deciduous oakwoods as a result of wood clearance for cultivation and pasture. Jahns [13] suggests that the reduction of forest cover by human activity favoured an expansion of *Pinus* and an increase of sclerophyllous plants. Very recently, Koskeridou and colleagues [29] have analysed some pollen samples from a core taken almost 1 km SW of our sequence, in a study where the main aim was to reconstruct the evolution of the

coastal wetland at Lerna. The absence of feasible radiocarbon dates, however, prevented authors from building an age-depth model. Local hygrophilous plants prevail over the surrounding vegetation, which is mainly represented by pines and deciduous oaks. Moreover, no major vegetation changes have been highlighted, apart from the increased presence of *Plantago lanceolata* in the uppermost levels suggesting anthropogenic disturbance.

Jahns' record from Lake Lerna has been widely used as one of the key pollen records from the Peloponnese and contributed to the reconstruction of human-environment dynamics in the Peloponnese in recent syntheses [1–3]. Other pollen records from the northeastern Peloponnese have been published by [27, 30]. In addition, a limited number of pollen records exist from other areas of the Peloponnese and adjacent regions (Fig 1A). In the study we will compare some of these data with our new record from Lake Lerna.

## Climate

The regional climate is typical of the eastern Mediterranean with hot, dry summers and mild, wet winters [31]. Most annual precipitation falls between November and March and the water balance (evapotranspiration/precipitation) is negative between April and October (Fig 1C). The mean annual precipitation is 383 mm and the mean annual temperature is 17˚C, with mean monthly temperatures ranging from 28˚C in July to 8˚C in January (data from Argos for the period 1980–2010; [32]).

The climate of the Peloponnese is influenced by Atlantic circulation patterns modulated by both Mediterranean and local air-mass systems [33]. The peninsula is located in the intermediate position between the high latitudes, dominated by the Siberian High, the North Atlantic Oscillation (NAO) and the North Sea–Caspian Pattern (NCP), and the low latitudes with the Subtropical High and the monsoonal systems [34]. Moreover, the mountain ridges stretching from North to South in the Peloponnese block westerly air masses and cause drier and relatively cooler conditions on the northeastern side of the peninsula [35]. Despite the relatively stable summer regime, extreme conditions and heavy rainfall events are occasionally induced by the incoming cool air from higher latitudes during anticyclonic episodes [36, 37].

A number of palaeoclimatic records are available from the Peloponnese, based on geochemical proxies from palaeolake sediments and cave speleothems (Fig 1A). The totality of the records from the Peloponnese outlines a complex, and not always consistent, climatological pattern that is likely due to a combination of chronological precision, geographical and topographical differences and the specifics of the sample environment. The complementarity of the different proxies is not yet fully understood and differences and similarities will be outlined in the following discussion, highlighting the local isotopic ($\delta D_{23}$) record from Lake Lerna [34] in relation to other Aegean and Eastern Mediterranean palaeoclimate records.

## Human occupation

The Argive Plain has been a key area for our understanding of ancient societies in Greece across the ages. The area was intensively occupied from at least the Early Bronze Age (for absolute chronologies, see Fig 1D) and different socio-economic developments through times are testified by some of the most notable archaeological sites in the region (Fig 1B). A number of settlements are known from the vicinity of ancient Lake Lerna, such as Lerna (ca. 3.5 km SW of the coring location) and Kephalari Magoula (ca. 2 km NW of the coring location and by the shore of ancient Lake Lerna), and Argos (ca. 6 km N of the coring location, by the Inachos River) which were all occupied during most of the Bronze Age [38]. The more well-known Late Bronze Age palatial centres at Mycenae and Tiryns occupied the eastern part of the plain. The Early and Late Bronze Age were periods of prosperity and high human activity, during

which the number of identified settlements increased also beyond the core areas, as evidenced by the results of the Berbati-Limnes Archaeological Survey conducted in an enclosed valley and uplands immediately adjacent to the Argive Plain [39]. On the contrary, the late Early Bronze Age and the Middle Bronze Age were marked by a reduction of the number of sites, but the Argive plain remained a centre of activity also during these periods. The end of the Late Bronze Age palatial period also coincided with a significant decline of human activity on the plain. From the end of the Early Iron Age, however, the city-state (*polis*) of Argos became increasingly dominant and by the early Classical period seems to have controlled the entire Argive Plain [40, 41]. In this period, the area again became densely occupied including a small settlement at Lerna itself [42]. In both the Hellenistic and Roman period, Argos continued to act as a significant regional centre and excavations carried out in the modern town have revealed extensive remains relating to urban activity in these periods [43–45]. During the Late Antiquity the importance of Argos is still confirmed by the intense building activity and evidence of human occupation is also attested through the plain [46–48]. From the seventh to the ninth century, often called the Byzantine Dark Ages, the archaeological findings seem to indicate a strong reduction in the population although Argos continued to be occupied [47]. After the re-establishment of Byzantine power in the Balkans, the significance of Argos and Nafplio as important commercial spots of the region increased [49, 50]. A picture of growth and prosperity is visible throughout the plain in this period: the agricultural expansion and intensification of the rural landscape is demonstrated by results of the Berbati-Limnes Archaeological Survey [51]. After the coming of the Latins and the reorganization of land in the Peloponnese during the Frankish and the first Venetian period, economic activities continued and the rural development intensified [49, 51, 52]. The sparsity of population information in Byzantine historical sources hinders us in following closely the demographic trends of the period [53]. In 1397 Yakub Pasha captured Argos and deported, according to a Venetian document [54], up to fourteen thousand–thirty thousand, according to a Byzantine historian [55]–of its inhabitants to Asia Minor. In 1463, the Argives entered into an agreement with Sultan Mehmed II, according to which they were deported to Istanbul [56]. The sources remain silent on the status of the town until 1479, when the Ottoman-Venetian treaty ratified Ottoman rule. These losses were counterbalanced during the *Pax Ottomana*. According to four detailed and hitherto unpublished Ottoman taxation cadasters of the sixteenth century, the Argive Plain was dominated by two urban centres (Argos and Nafplio) and contained 10 identified villages and 50 farms, whose population reached its peak in the early 1570s. The earliest proper population census and cadastral survey of the area was carried out by the Venetian authorities in 1700 and attests a noteworthy demographic growth and the emergence of new settlements [57–59]. Right after the Greek War of Independence, the *Expédition scientifique de Morée* conducted a survey of the entire Peloponnese in 1829 [60], which was then followed by regular censuses commissioned by the nascent Greek state. On the basis of these records, we observe demographic stabilisation in the first half of the nineteenth century and increase in the second one [61].

## Materials and methods

### Sediment sequence and age-depth model

The sediment core obtained from the ancient lake (37˚34'57.66" N, 22˚43'57.60" E, 1 m a.s.l.) has a total length of 5 m. The coring was performed using a hand-operated Russian peat corer. The sequence was divided into lithological units on the basis of changes in colour, boundaries and internal structure of sediments (Fig 2) and the amount of organic content (TOC). More details about the coring and the TOC analysis can be found in [34].

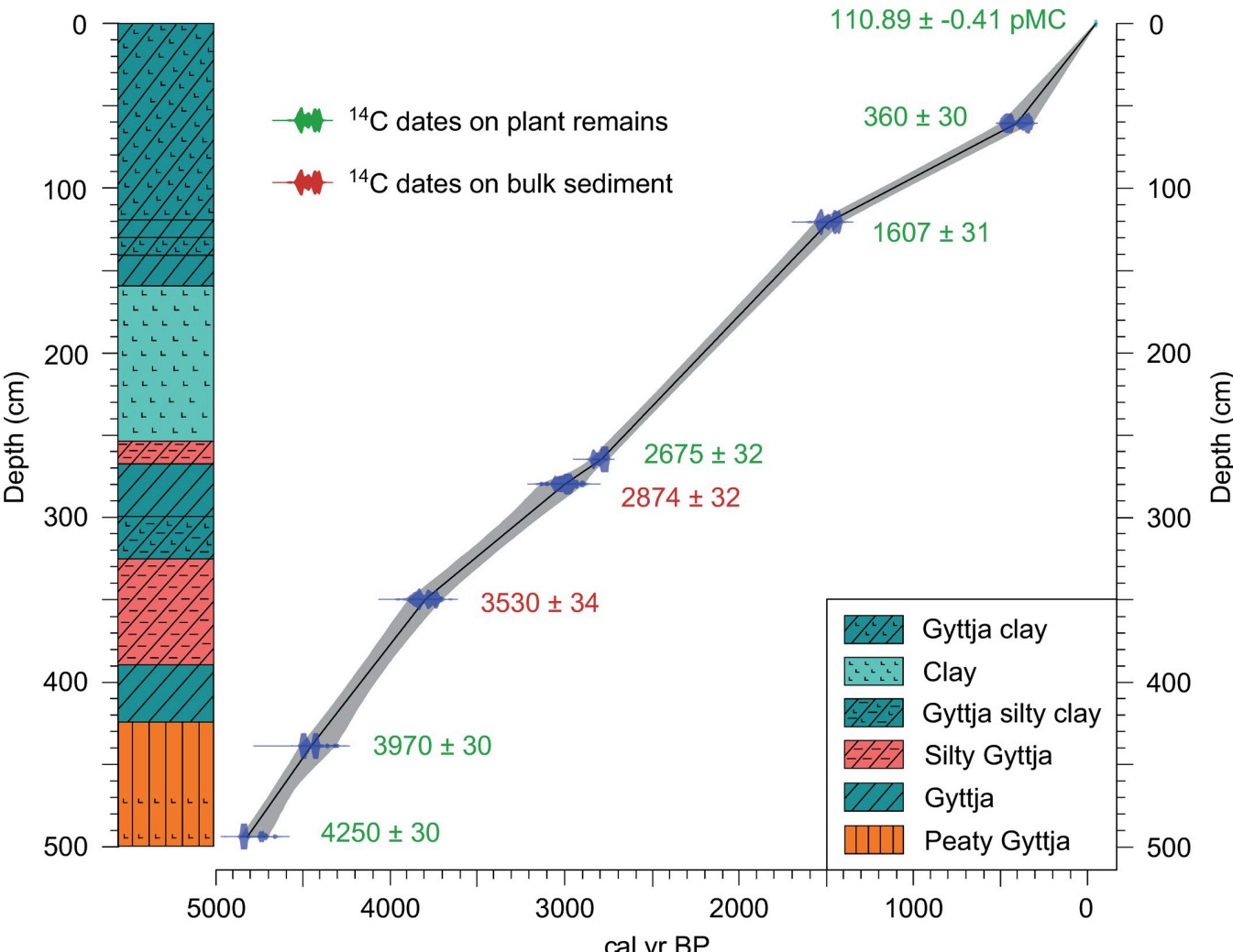

**Fig 2. Sedimentological and chronological sequence of the study core from ancient Lake Lerna (Peloponnese, Greece).** (left) Sediment succession and (right) age-depth model, built with Clam 4.1.1 R-package (linear interpolation; [63]) and based on 8 radiocarbon ages (calibrated with IntCal20; [62]). Results of radiocarbon dating have been previously published by [34].

The core covers the last 5000 years. The age-depth model was previously published by [34] and was based on 6 AMS radiocarbon dates of plant remains and 2 dates of organic-rich sediment. For this study it has been updated using the new calibration curve IntCal20 [62]. The new age-depth model has been provided using the Clam R package (v4.1.1) [63] and it was used for all data presented in the study (Fig 2).

## Pollen analysis

The analysis of pollen grains and Non-Pollen Palynomorphs (NPPs) has been carried out on 84 sediment samples. Among them, a total of 64 samples have been included in the results, taking into consideration the minimum count per sample of 200 pollen grains and 50 NPPs. The mean sampling resolution is 6 cm and the mean chronological resolution is 76 years. A known amount of dry sediment (2.8–14 g) for each sample was chemically processed with alternating treatment of HCl (37%), HF (40%) and hot NaOH (10%) ([64], modified) in order to remove

mineral and organic matter from the sediments. *Lycopodium* spores were added to estimate pollen, NPP and charcoal concentrations [65]. The identification of pollen morphology is based on identification literature [64, 66–70] and reference pollen collections. Following [71], pollen grains of *Quercus* species are grouped into: *Quercus robur* type including all deciduous oaks, *Quercus cerris* type grouping the semi-evergreen oaks plus the evergreen *Q. suber* and *Quercus ilex* type where only evergreen species are included. *Pinus* refers both to the Mediterranean and montane species. Among Poaceae, cereals have been identified following [64, 72] and they are split in *Avena/Triticum*, *Hordeum* group and *Secale*, besides those pollen grains whose deterioration prevented the identification (Cerealia type). Cichorieae include only pollen of the European tribe of Asteraceae with fenestrate pollen grains [73]. The identification of NPPs is based on specific publications ([74–86]; for the acronyms see [87]).

The percentage values have been calculated on different pollen sums depending on the group. The basic pollen sum includes all the terrestrial spermatophytes with the exception of Cyperaceae. Other pollen, spore, and NPP percentages (Cyperaceae, macrophytes or aquatic plants, indeterminable grains, ferns, green algae, fungi, and all the others) have been calculated as the sums of terrestrial spermatophytes and, in turn, each group of considered palynomorphs [88]. The concentration values have been calculated per weight unit of sediment (pollen grains-NPP remains/g; hence p-NPP/g). The influx values have been obtained from the concentration values on the basis of the sedimentation rates as inferred from the age-depth model (pollen grains-NPP remains/cm$^2$ year; hence p-NPP/cm2 y; [88]). Pollen influx data (i.e. the amount of pollen grains incorporated annually per unit of sediment) is an estimation of the plant biomass.

The pollen diagrams have been plotted against depth and time scales using the TILIA program [89]. All Arboreal Pollen (AP) and Non-Arboreal Pollen (NAP) taxa with values >2% of the basic pollen sum have been considered in the CONISS cluster analysis [89]. The OJC (*Castanea*, *Juglans*, *Olea*) group is based on [90], API (Anthropogenic Pollen Indicators: *Artemisia*, *Centaurea*, Cerealia type, *Avena/Triticum*, *Hordeum* group, *Secale*, Cichorieae, *Plantago* undiff., *Plantago lanceolata* type, *Urtica*) follows [91] and PI (Pastoral Indicators: Asteroideae, *Galium*, Ranunculaceae) is adapted from [92].

With the aim to study past fire activity charcoal particles have been identified and counted in the pollen slides. The charcoal particles have been sorted in three dimensional classes according to the size of their shortest axis. Interpretation of the charcoal data is according to [93, 94] where fragments of 10–50 μm represent regional fire, fragments of 50–125 μm indicate fire occurred at landscape/regional scale and fragments of more than 125 μm represent evidence of local fire. Charcoal influx data (charcoals/cm$^2$ year, hence c/cm2 y) is based on concentration values (charcoals/g; hence c/g) and is an estimate of the burnt biomass.

## Results

Pollen grain preservation is variable and the number of indeterminable grains (broken or degraded) spans from less than 1% to 18%. The mean count of terrestrial spermatophytes, Cyperaceae excluded, is 215 pollen grains/sample. Pollen concentrations are low and range from 421 (at 94 cm—990 cal yr BP) to 34,400 (at 132 cm—1580 cal yr BP) p/g. A total of 125 pollen taxa has been identified, including 48 arboreal, 57 herbaceous and 20 aquatic taxa. NAP prevails up to 84% of the pollen sum. Percentage, concentration, and influx values of selected taxa are plotted in Figs 3 and 4.

The mean count of NPPs is 939. Among 65 taxa recognized, a selection based on their ecological value is provided in Figs 5 and 6. Concentrations of selected NPPs vary between 18 (at 94 cm—990 cal yr BP) and 11,200 (at 21 cm—110 cal yr BP) NPP/g.

## LAKE LERNA (1 m a.s.l.) - Peloponnese, Greece
Pollen percentage diagram

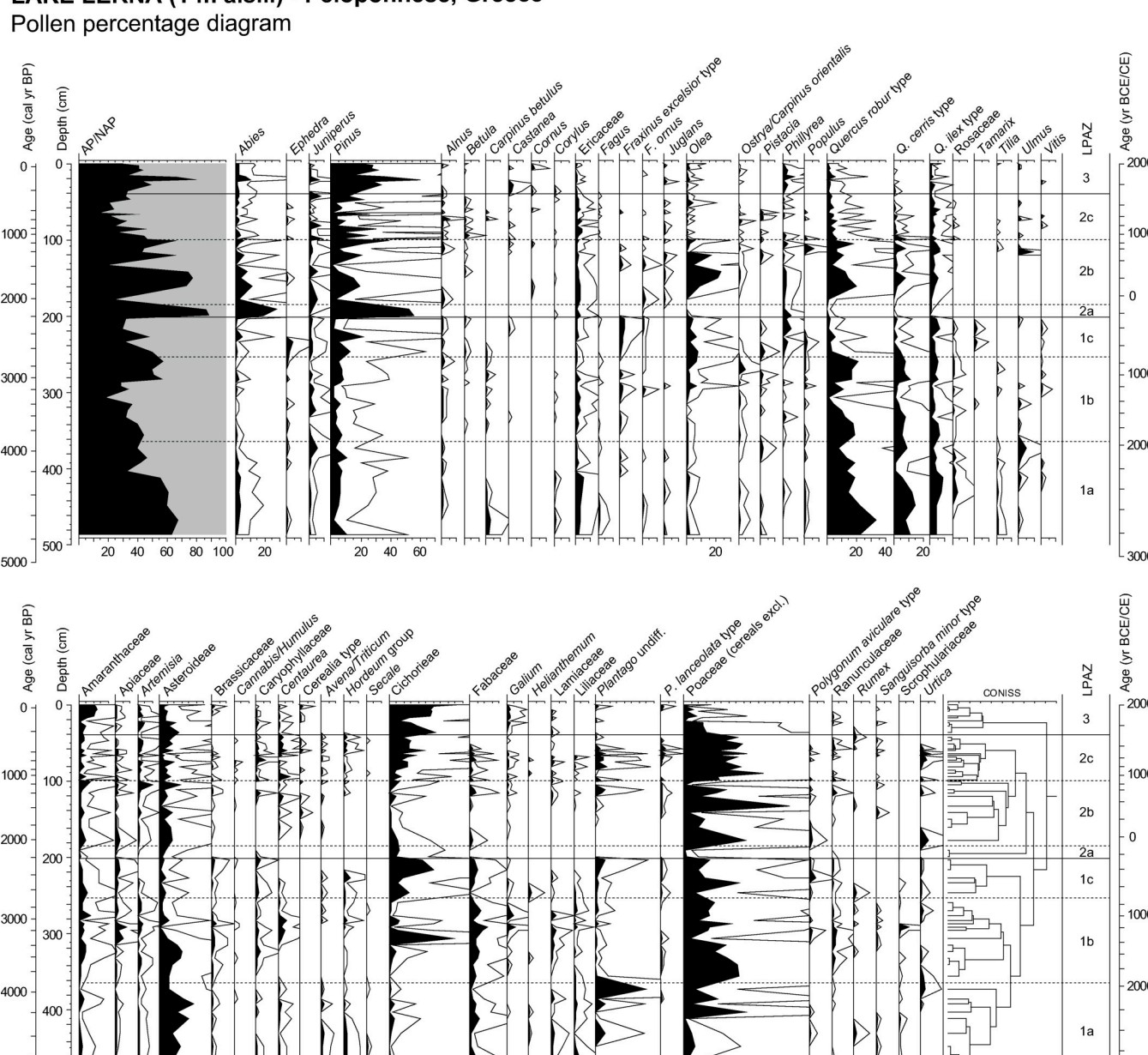

Cristiano Vignola & Nichola A. Strandberg 2019-2020

**Fig 3. Results of pollen analysis of the study core from ancient Lake Lerna (Peloponnese, Greece).** Pollen percentage diagram of arboreal and non-arboreal selected taxa. Curve magnification 4x.

Seven pollen zones and sub-zones, named from LPAZ 1a (bottom) to LPAZ 3 (top), are described. Pollen zonation was determined by CONISS cluster analysis and visual inspection. Ages are expressed in calendar years BP and the estimation is rounded up to the nearest 10 years.

**LAKE LERNA (1 m a.s.l.) - Peloponnese, Greece**

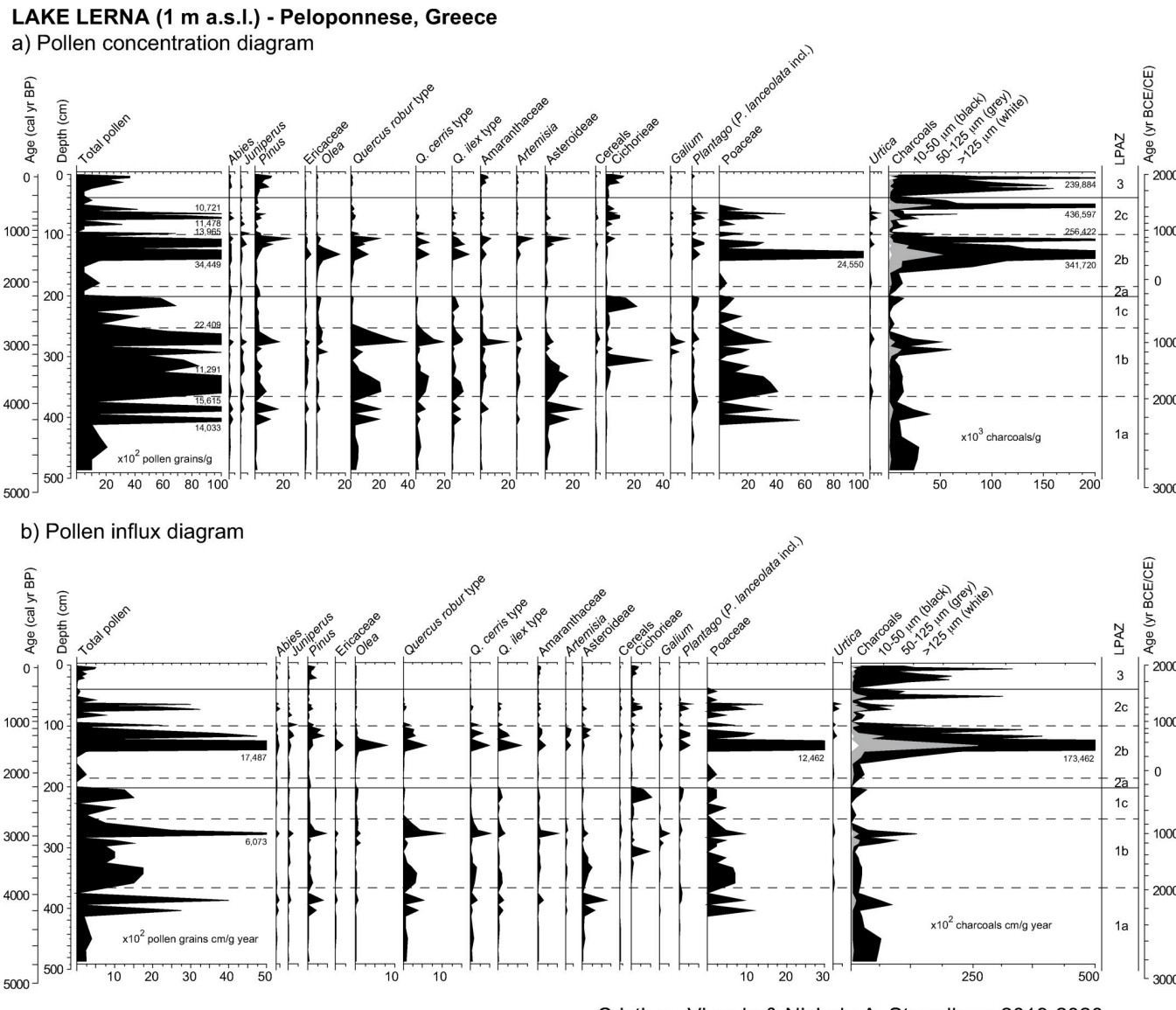

**Fig 4. Results of pollen analysis of the study core from ancient Lake Lerna (Peloponnese, Greece).** (a) Pollen concentration and (b) influx diagrams of selected arboreal and non-arboreal taxa.

## LPAZ 1: 486–202 cm, ca. 4760–2240 BP (2810–290 BCE)

**LPAZ 1a: 486–374 cm, ca. 4760–3980 BP (2810–2030 BCE).** The sub-zone corresponds to the Early Bronze Age and is characterised by the progressive decrease of AP. Concentration is low but increases up to 15,600 p/g in the second part. The number of pollen taxa varies from 29 to 41. Arboreal taxa are dominated by *Quercus robur* type (8–34%) and *Quercus cerris* type (2–15%). *Quercus ilex* type (2–8%) and Ericaceae (0–5%) have the highest values among Mediterranean trees and shrubs, whereas *Abies* (max 4%) and *Fagus* (max 2%) represent montane vegetation. *Pinus* is present but never exceeds 10%. Among arboreal synanthropic taxa, *Olea* is present from the bottom sample with low percentage values (0–2%), whereas pollen grains of *Juglans* and *Vitis* are sporadic (max 1%). Herbs are dominated by Asteroideae (6–23%) and

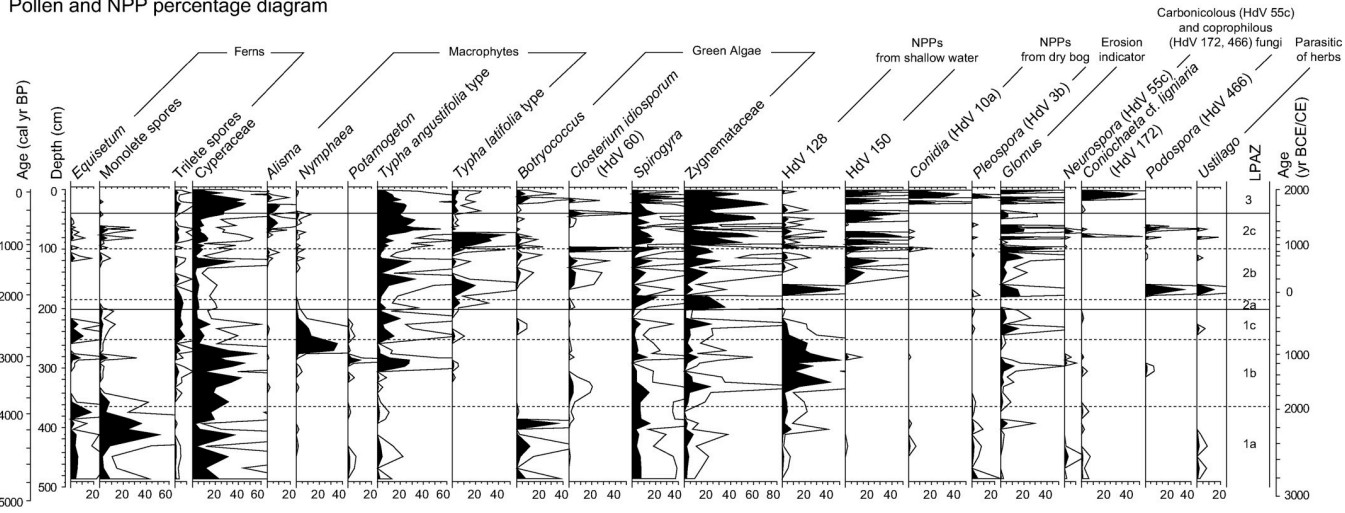

**Fig 5. Results of pollen analysis of the study core from ancient Lake Lerna (Peloponnese, Greece).** Pollen percentage diagram of selected aquatic and NPP taxa. Curve magnification 4x.

Poaceae (1–23%). Cereal pollen is present from the bottom sample and shows a decreasing trend (*Avena/Triticum*: max 2%, *Hordeum* group: max 2%, *Secale*: max 1%). At the end of the sub-zone *Plantago* undiff. has a peak reaching the highest value of the core (35%).

Macrophytes are also present, among these *Typha angustifolia* type (max 5%) prevails. Nonetheless the sub-zone is characterised by the highest presence of ferns within the core (monolete spores, 2–52%) and by green algae, among which *Botryococcus* has the highest value (35%). Among NPPs, *Pleospora* and *Ustilago* (max 2%) are present; a peak of *Glomus* (6%) is attested at ca. 4120 BP. Concentration and influx values of charcoal fragments are quite low (7,800–42,500 c/g and 1,700–9,0002 c/cm$^2$ y respectively) and the fraction of fragments larger than 125 μm is absent.

**LPAZ 1b: 373–259 cm, ca. 3980–2740 BP (2030–790 BCE).** This sub-zone corresponds to the beginning of the Middle Bronze Age until the Early Iron Age. The decrease of AP continues, reaching the second lowest value of the sequence (19%) at ca. 3310 BP. After that, AP values increase again. The pollen concentration and influx vary (1,200–22,400 p/g and 233–6,100 p/cm$^2$ y respectively), with a significant increase at ca. 2950 BP. The identified terrestrial spermatophytes count from 23 to 46 taxa. *Quercus robur* type (max 22%) and *Quercus cerris* type (max 12%) remain the predominant taxa. Apart from *Quercus ilex* type (1–7%), the Mediterranean vegetation is represented by shrubs such as *Juniperus* (max 4%), *Phillyrea* (max 2%) and *Pistacia* (max 2%). *Olea* (max 7%) and *Vitis* (max 2%) slightly increase in the second part of the sub-zone. NAP is mostly dominated by Poaceae (4–37%) while Cichorieae have an abrupt peak (44%) at ca. 3310 BP. Cereals almost disappear in the first part of the sub-zone, whereas in the second part the presence of *Hordeum* group (max 3%) is accompanied by other herbaceous synanthropic taxa (Apiaceae, *Centaurea*, *Polygonum aviculare* type, *Sanguisorba minor* type).

Among macrophytes *Typha angustifolia* type increases up to 28%, as well as *Potamogeton* (11%) and, especially at the end of the sub-zone, *Nymphea* (36%). Ferns reduce significantly, in contrast to green algae. *Glomus* (0–11%) also increases after ca. 3310 BP, even if the most abundant NPP of this sub-zone is HdV-128 (2–50%). Concentration and influx values of

### LAKE LERNA (1 m a.s.l.) - Peloponnese, Greece
a) Pollen and NPP concentration diagram

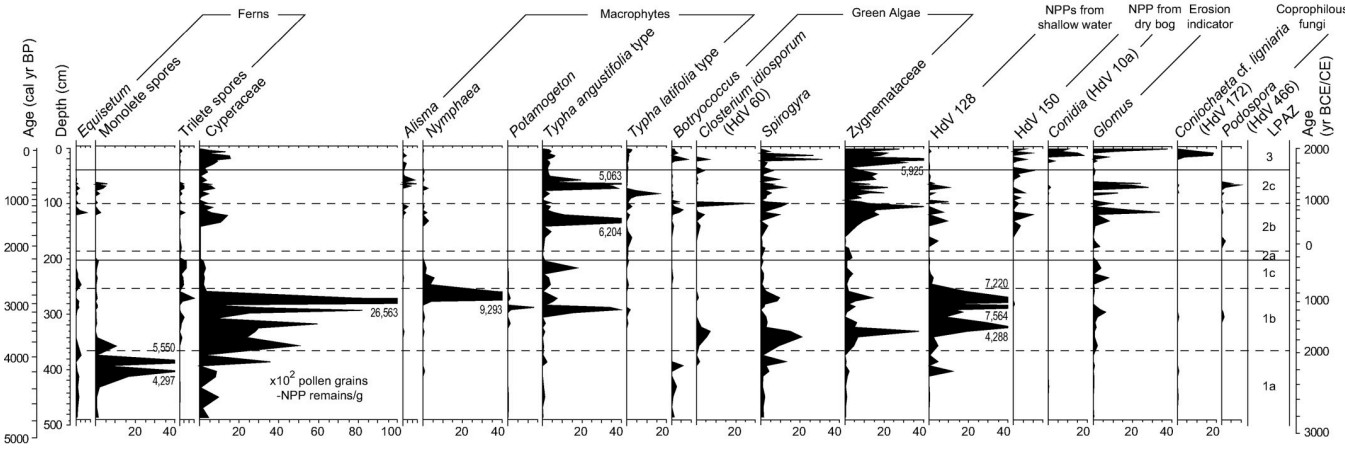

b) Pollen and NPP influx diagram

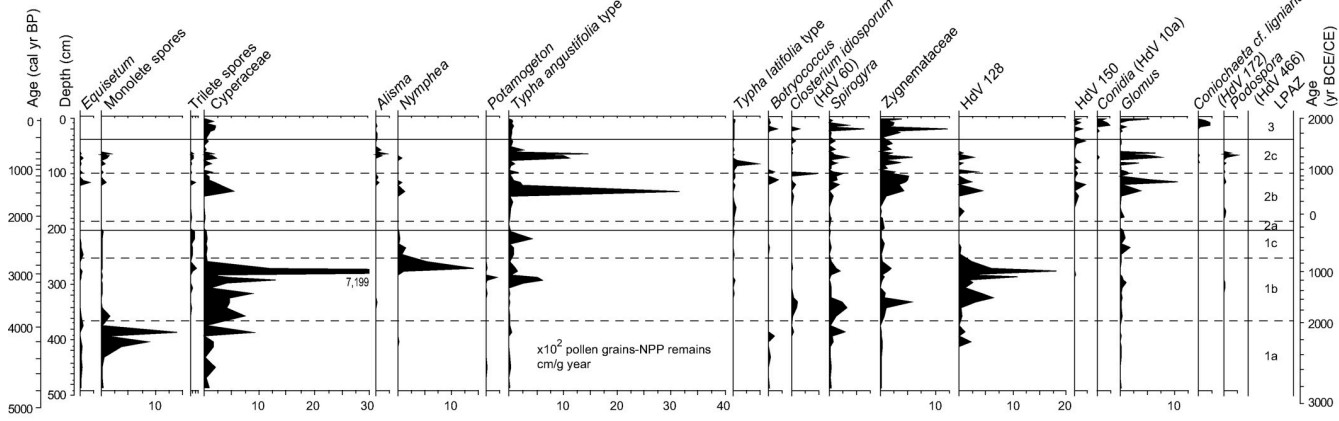

Cristiano Vignola & Nichola A. Strandberg 2019-2020

**Fig 6. Results of pollen analysis of the study core from ancient Lake Lerna (Peloponnese, Greece).** (a) Pollen concentration and (b) influx diagrams of selected aquatic and NPP taxa and charcoals.

charcoal remain low in the lower part of the sub-zone (434–72,700 c/g and 63–15,400 c/cm$^2$ y respectively) and increase after 3310 BP, when the fraction larger than 125 μm is also present.

**LPAZ 1c: 258–204 cm, ca. 2740–2240 BP (790–290 BCE).** The sub-zone covers the end of the Early Iron Age to the Early Hellenistic period and is characterised by a rapid decrease of AP (up to 30%). The pollen concentration, as influx, is very low but increases up to 7,100 p/g after ca. 2450 BP. The number of terrestrial spermatophytes varies from 31 to 47. Pollen from thermophilous trees almost disappears, whereas Mediterranean forest taxa increase slightly thanks to the enhanced presence of *Phillyrea* (max 6%) together with *Quercus ilex* type (max 6%). The montane taxa are mostly represented by *Abies* (max 5%) and *Betula* (max 1%), whereas *Pinus* increases up to 22% at ca. 2450 BP. Among cultivated trees, only *Olea* reaches values above 2%. The appearance of *Tamarix* (0–2%) among riverine plants is noteworthy. The predominant herbaceous taxa are Poaceae (10–37%) and, after ca. 2450 cal yr BP, also Cichorieae (up to 31%). *Plantago* (1–7%) mainly accompanies the frequency of Cichorieae. Cereal pollen disappears after ca. 2450 BP.

*Typha angustifolia* type prevails (2–19%), whereas *Nymphea* decreases markedly and reaches only 1%. Fern spores are widely present while green algae almost disappear after ca. 2450 BP; among fungi, *Glomus* prevails after ca. 2620 BP (1–15%). The smallest charcoal fragments have low concentration and influx values (764–14,200 c/g and 154–3300 c/cm$^2$ y respectively) but increase after ca. 2450 BP. The amount of indeterminable pollen grains reaches the highest value of the diagram (18%).

### LPAZ 2: 203–42 cm, ca. 2240–270 BP (290 BCE-1680 CE)

**LPAZ 2a: 203–191 cm, ca. 2240–2120 BP (290–170 BCE).**   The sub-zone is represented by only two sediment samples and shows the highest AP percentage values (87–89%) together with some of the lowest pollen concentration values (571–646 p/g) of the diagram. The total pollen influx tends to zero. The number of pollen taxa is the lowest of the diagram (14). Arboreal taxa are mainly represented by *Pinus* (53–55%) and *Abies* (20–28%), the herb ones by Cichorieae (max 8%). Macrophytes percentage values are low (max 7%), in contrast to the amount of trilete spores (6–7%) and a peak of green algae (Zygospores: 35%). No fungal remains are present. Charcoal concentration and influx tend to zero and the number of indeterminable pollen grains reaches the lowest value of the diagram (<1%).

**LPAZ 2b: 190–101 cm, ca. 2120–1130 BP (170 BCE-820 CE).**   The sub-zone, representing the Late Hellenistic to the Early Byzantine period, is characterized by AP vs. NAP oscillations. Total pollen concentration and influx (34,400 p/g and 126–17,500 p/cm$^2$ y respectively) have the highest peaks of the diagram at ca. 1580 BP. The number of identified terrestrial spermatophytes are 23–38 taxa. The most abundant arboreal taxon is *Pinus* (1–40%). Although *Quercus robur* type (6–20%) and *Quercus cerris* type (1–9%) expand, the number of thermophilous taxa reduces. *Quercus ilex* type and *Juniperus* prevail among Mediterranean taxa (reaching both 6%), whereas *Betula* (max 1%) is present among montane trees in the second part of the sub-zone. Between ca. 1840 and 1480 BP *Olea* reaches the highest value of 23% and *Juglans* appears (max 2%). NAP shows the dominance of Poaceae reaching the highest value (71%). Among other herbs, cereals and synanthropic taxa are mostly represented: *Artemisia*, Asteroideae, *Centaurea*, *Galium*, *Plantago* undiff., *Plantago lanceolata* type and Ranunculaceae (max 15%).

Hydrophilous plants also increase between ca. 1840 and 1480 BP (*Typha angustifolia* type: 3–33%, *Typha latifolia* type: 0–20%). From bottom to top level ferns reduce, green algae increase and *Glomus* oscillates (0–19%). Among the other NPPs, HdV-150 (0–27%) and HdV-128 (0–50%) prevail while *Podospora* (33%) and *Ustilago* (14%) have a peak at ca. 1920 BP. Concentration and influx values of charcoals abruptly increase in the second part of the sub-zone (10,300–395,300 c/g and 2,100–200,700 c/cm$^2$ y respectively): the fraction 10–50 μm reaches the highest peaks and the larger one increases.

**LPAZ -2c: 100–42 cm, ca. 1130–270 BP (820–1680 CE).**   The decrease of AP up to the lowest percentage value (16%) characterises the sub-zone spanning from the Late Middle Ages to the Ottoman period. The pollen concentration spans from 442 p/g at ca. 990 BP (i.e. the lowest value of the diagram) to 11,500 p/g in the succeeding level. The number of terrestrial spermatophytes varies from 22 to 42. *Pinus* shows significant oscillations (1–31%) and thermophilous taxa decrease (*Quercus robur* type: max 7%, *Quercus cerris* type: max 3%). Worth mentioning is the expansion of *Betula* (max 4%) and the presence of *Fagus* (max 1%) among the montane vegetation. *Olea* strongly decreases but *Castanea* and *Juglans* are now attested. NAP is dominated by Poaceae (21–54%) while Cichorieae values increase within the sub-zone (3–24%).

Among macrophytes the increase of *Typha angustifolia* type (max 55%) and *Typha latifolia* type (max 44%) characterizes the first part of the sub-zone, whereas *Alisma* (max 9%) expands

in the second part. Apart from the continuous increase of green algae, *Glomus* presents oscillating peaks (max 44%) and *Podospora* (max 11%) is attested until ca. 450 BP. HdV-150 is the most abundant NPP. In the sub-zone the highest peak in concentration of the smallest charcoal fragments (436,600 c/g) is found at ca. 330 BP; the fraction 50–125 μm also increases.

### LPAZ 3: 41–0 cm, ca. 270 BP-present

The uppermost pollen zone shows increasing AP values until ca. 110 BP. Total pollen concentration and influx are low (702–4,000 p/g and 89–609 p/cm$^2$ y respectively) and the total number of pollen taxa varies from 17 to 30. *Pinus* reaches the highest percentage value of the diagram (60%). Thermophilous taxa progressively decrease up to 2% and Mediterranean shrubs expand (*Phillyrea*: max 9%, *Juniperus*: max 7%). NAP is mainly characterised by the exponential growth of Cichorieae (8–48%).

The reduction of *Typha angustifolia* type (1–25%), *Alisma* (0–11%) and *Typha latifolia* type (0–5%) among hydrophilous taxa is accompanied by the exponential increase of *Glomus* (0–48%), *Coniochaeta* cf. *ligniaria* (0–35%) and *Conidia* (0–42%). The concentration and influx values of charcoal fragments constantly increase (7,200–254,100 c/g and 906–35,200 c/cm$^2$ y respectively) and the highest peaks of the fraction larger than 125 μm are found.

## Interpretation and discussion

### The Early Bronze Age (5150–4000 BP)

The Lerna sequence starts during the mid-Holocene with a prevalence of pollen indicating thermophilous vegetation, representing the mixed deciduous oak woodland of the Peloponnesian uplands (Fig 7). It was dominated by deciduous *Quercus* together with other thermophilous trees until ca. 4300 BP. The high degree of forest cover might be related to increased moisture as also testified in the Mavri Trypa record from 4700 to 4400 BP [95] as well as in the Aegean Sea for the period 5400–4300 BP [96, 97] and in the southern Mediterranean for 5000–4600 BP [98]. This wet phase interrupted the general aridity trend reported for the whole Balkans, based on a compilation of several palaeoclimate records from the Balkan peninsula and the Aegean Sea [99]. Pollen of *Quercus ilex* type and Ericaceae represents vegetation of the lowlands, where the open Mediterranean forest in association with shrublands was widespread. Montane taxa are present (*Abies*, *Fagus*) and refer to the altitudinal forest, which nowadays is spread on the highest mountains to the north and east of the study area and is mainly composed of *Abies cephalonica* and *Pinus nigra*. Notably, the Peloponnese is outside the present-day natural distribution of *Fagus*, whose findings are sporadic in our core (Fig 3).

The lithological sequence shows peaty deposits at the very bottom of the core combined with the highest TOC values (Figs 2 and 7). At the same time, high sedimentation rate is inferred from the age-depth model and low pollen concentrations are attested. Although the presence of *Botryococcus* remains indicates depositional settings affected by freshwater [100], the contemporary scarce amount of other aquatic plants and the abundance of Cyperaceae shows that this part of the basin was a fen with marsh vegetation (Fig 4). The same environmental conditions are attested ca. 1 km SW of our coring site by paleoenvironmental data of [29] but differ from the vegetation reconstruction provided by the previous study of Jahns ([13] and Fig 8). This suggests local conditions which were not widespread throughout the ancient lake. Nowadays a fen is present in the study area.

Cereals are present in high amounts from the bottom to ca. 4190 BP (Fig 7). They are low pollen-producers and their abundance testifies the wide extent of cultivated fields in the plain, probably pertaining to nearby ancient settlements, such as the Early Bronze Age settlements of Kephalari Magoula, situated by the shore of Lake Lerna, and Argos along the Inachos River

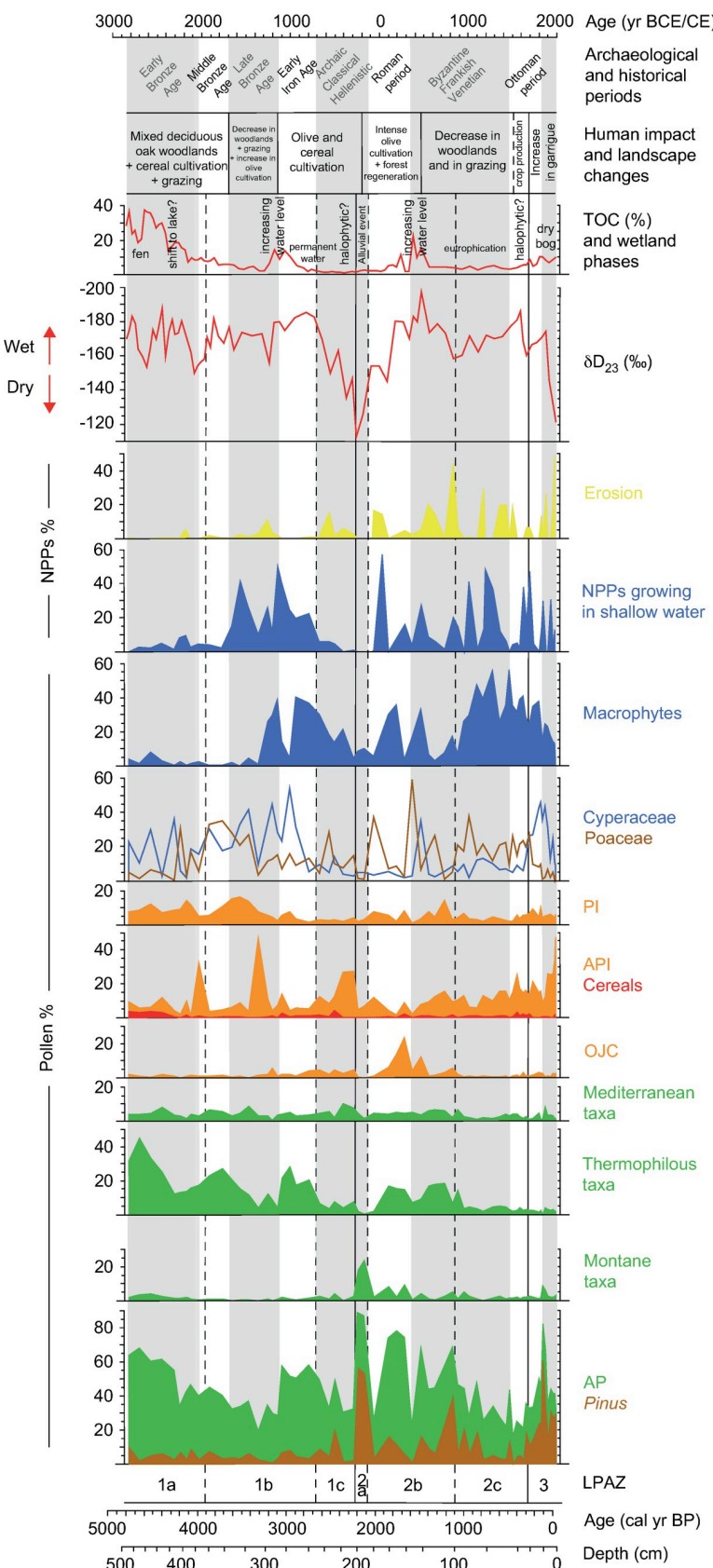

**Fig 7. Comparison between pollen and geochemical data of the study core from ancient Lerna, Peloponnese (Greece).** Pollen percentage diagram of ecological groups and other indicators (OJC, API, PI; see §3.2), plotted against age and aligned with geochemical data [34]. For more details on the stable hydrogen isotope analysis refer to the original publication. Montane: *Abies*, *Betula*, *Fagus*, *Picea*; Thermophilous: *Buxus*, *Carpinus betulus*, *Corylus*, *Fraxinus excelsior* type, *Hedera*, *Ostrya/Carpinus orientalis*, *Quercus robur* type, *Quercus cerris* type, *Tilia*, *Ulmus*; Mediterranean: *Cistus*, *Fraxinus ornus*, *Pistacia*, *Phillyrea*, *Quercus ilex* type, *Rhamnus*. Wetland phases at the coring site and landscape changes in the Argive Plain are summarised here.

[101]. An overall high human pressure in the Argive Plain is testified by archaeological evidence confirming the growth of socio-economic complexity during the Early Bronze Age II period (ca. 2650–2200 BCE/4600-4150 BP), not only at nearby Lerna and Tiryns but across the whole of southern Greece [38]. Other primary anthropogenic taxa, including *Olea* and *Juglans*, attest the role of cultivation in the local economy (see the archaeobotanical assemblage at Lerna in [102, 103]), confirming the early steps of management and cultivation in the Peloponnese that are argued to have started in the Early Bronze Age (from ca. 5200 BP) [104–106]. In this framework, the low frequency of *Olea* pollen as already attested by [13] (Fig 8), compared to other records of the Peloponnese (e.g. Kleonai; [30]) and Attica (e.g. Elefsis Bay; [107, 108]) (Fig 9), could be tied to the scarcity of available lands for olive growth since our record shows that at least parts of the Argive Plain was mainly used for the cultivation of cereals at that time (Fig 3). It is worth noting that a similar low amount of olive pollen grains has been recorded at Lake Lerna in the coring site of [29]. Coprophilous and carbonicolous fungi together with charcoal fragments indicate strong human pressure. Moreover, the presence of crop fields is confirmed by the occurrence of *Ustilago*, a parasitic fungus connected to cereals that may have benefitted from the wet environment [109–111] (Fig 5).

From ca. 4190 to 3980 BP, corresponding to the final centuries of the Early Bronze Age, a new phase of sedimentation in this part of the basin is attested by the deposition of gyttja sediments (Fig 2), and the pollen and NPP concentration and influx fluctuate (Figs 4 and 6). The stable hydrogen isotope (δD$_{23}$) record from the Lake Lerna sediments reveals that the Argive

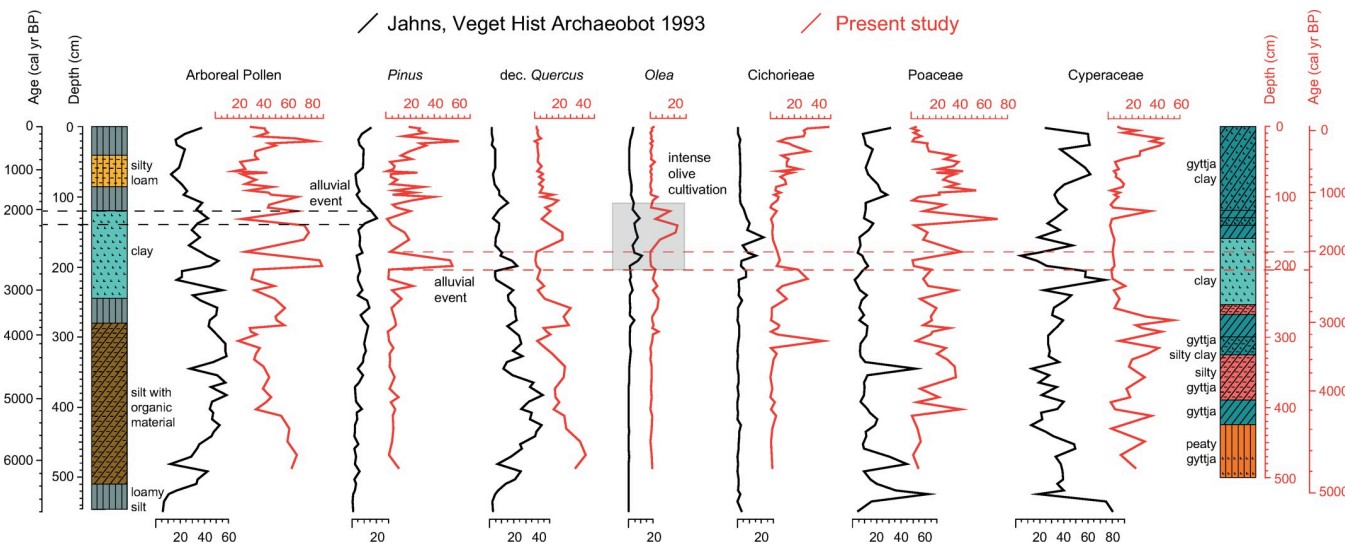

**Fig 8. Comparison between pollen data of the two sediment cores available from ancient Lerna, Peloponnese (Greece).** Pollen percentage diagram of selected taxa from the present study core (red) and from [13] (black), plotted against depth and aligned with sediment data. The dashed lines mark the alluvial event testified by peaks of *Pinus* pollen (and *Abies*) in both cores. The grey square shows the portion of the sequences where differences in *Olea* peaks suggest a different sedimentation rate between the two cores.

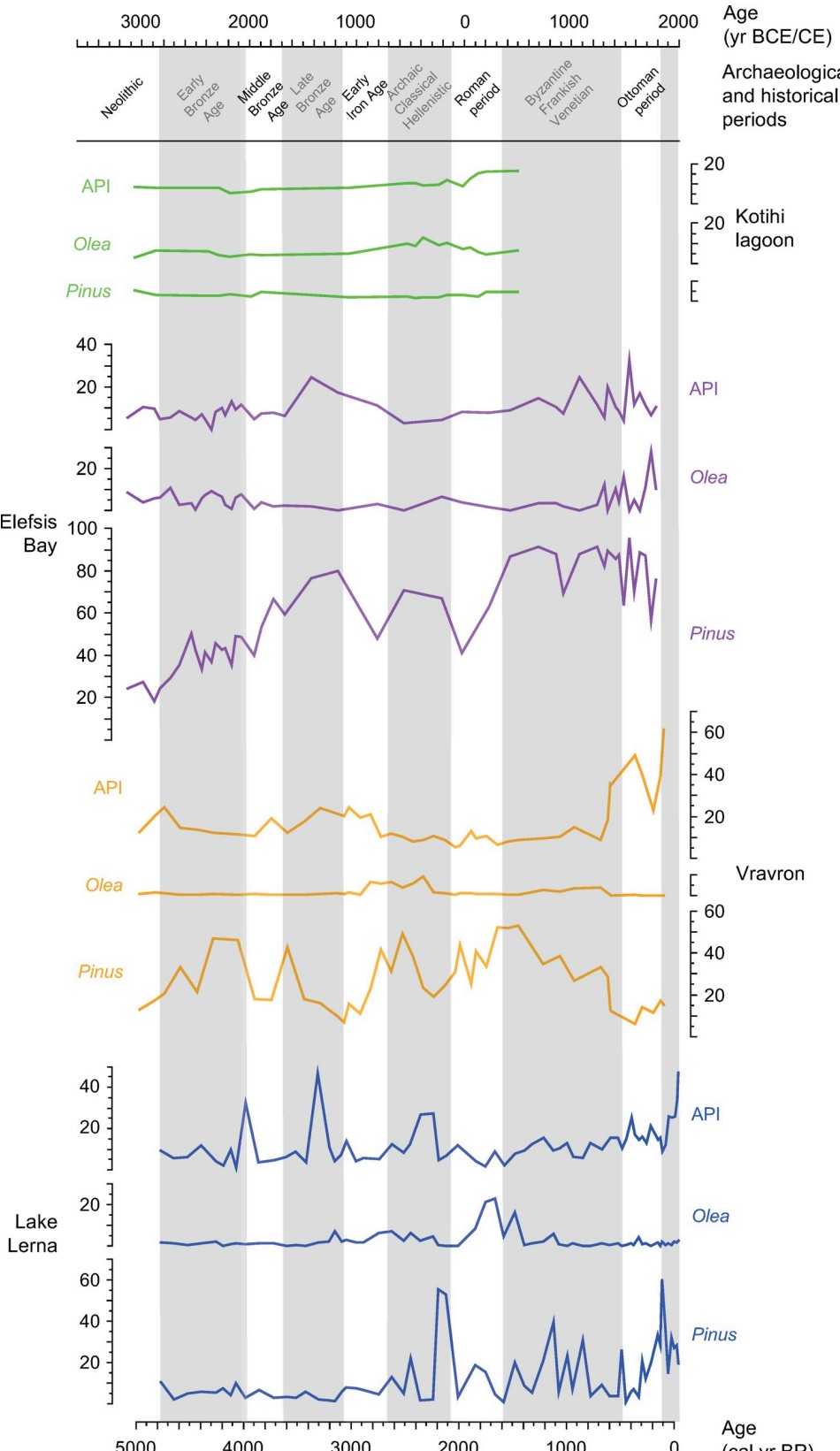

**Fig 9. Comparison between pollen sequences from the Peloponnese and southern Greece.** Pollen percentage diagram of selected taxa from ancient Lake Lerna (present study), Vravron wetland [2, 130, 188], Elefsis Bay [96, 107], Kotihi lagoon [2, 121]. For API group see §3.2.

Plain was characterised by a variable precipitation regime from ca. 4600 to 3900 BP, with a lasting dry phase that started at ca. 4120 BP and correlated with the onset of an arid phase in the Elefsis Bay (Saronic Gulf; [96]) (Fig 7; details about methods of the isotopic analysis can be found in [34]). Dry phases are also identified in the stable oxygen isotope records from Mavri Trypa Cave (from ca. 4400 BP) and Alepotrypa Cave (from ca. 4200 BP) [95, 112]. Many records from the Eastern Mediterranean generally show more arid conditions in the interval from approx. 4300 to 3800 BP although there are local variations [113]. Despite the evidence of lower precipitation during this time period our sediment record indicates a shift to deeper water conditions at our coring site, as seen by decrease of peaty elements in the gyttja. The pollen deposition increases, ferns almost disappear after reaching a peak in concentration and sedge vegetation starts to expand as shown by influx values. All this evidence seems to suggest that there was a vegetation shift at the coring site linked to a progressively increased water table. Species of Poaceae, however, can grow in different environments and could represent either a lacustrine vegetation belt of *Phragmites* (common reed) and other hygrophilous grasses or dry meadows and grasslands (Fig 7). In this portion of the sequence Poaceae are in agreement with the increase of Asteroideae as well as of *Plantago*, a group of herbs characteristic of footpath and ruderal communities [114] whose pollen reaches a peak at ca. 3980 BP (2030 BCE) (Fig 3). The counterintuitive nature of an expanding lake during drier climatic conditions seems to indicate human management.

The settlements of Lerna and Kephalari Magoula continued to be occupied during this period but the values of cereal pollen decrease suggesting a contraction of cereal fields in the vicinity of the coring site. A reduction of activity coincides with a general reduction of site numbers in the Peloponnese from the late Early Bronze Age [3]. Settlements remained generally small in size, and it is reasonable to suppose that the Argive Plain was characterized by less intensive land use, mainly focused on grazing activities, as attested by the peak of API represented by *Plantago*.

## The Middle and Late Bronze Age (4000–3150 BP)

An increase of humidity is visible in the $\delta D_{23}$ record from ca. 3900 BP. Speleothem data of the Peloponnese also indicate a wetter phase from the later phases of the Middle Bronze Age, as evidenced in the Mavri Trypa Cave (ca. 3800–3550 BP; [95]) and in Alepotrypa Cave (3900–3700 BP; [112]). In the pollen record Cyperaceae increase again starting from ca. 3860 BP while thermophilous trees prevail in the uplands until ca. 3720 BP. Despite these wetter conditions, there is an overall decline in AP at the same time as Poaceae and PI increase until ca. 3600 BP. This indicates that pasturelands expanded at the expense of the Mediterranean forest in the lowlands shaping the landscape of the Argive Plain starting from the Middle Bronze Age (Fig 7).

The first half of the Late Bronze Age, from ca. 3600 to 3310 BP (corresponding to 1650–1360 BCE), is characterised by wetter conditions both locally in Lerna as indicated by $\delta D_{23}$ record (especially ca. 3400–3200 BP) and in the Peloponnese, North Aegean, and Anatolia [95, 115–117]. Despite these generally wetter conditions, upland thermophilous vegetation declines significantly, accompanied by the increase of Mediterranean taxa in the lowlands (Fig 7). A clear decline in AP is generally attested in southern Greece from 3600 to 3400 BP [3]. The expansion of Mediterranean shrubs at the expense of thermophilous woodlands, as a result of

the intensive use of the landscape for economic activities, is also indicated by other palynological records from the Peloponnese (e.g. Akovitika, Aliki lagoon, Kiladha Bay, Kleonai, Kotihi lagoon; [30, 118–121] as well as from Messolonghi in the Gulf of Corinth (e.g. Klisova lagoon; [122]). These changes correlate with a period of socio-economic expansion in the Argive Plain during the Late Bronze Age, with significant activities at Lerna and Argos and with the emerging palatial centres of Mycenae and Tiryns taking a leading role [123, 124]. Apart from the slight increase of cereal pollen, the highest percentage values of PI confirm the role of human communities in opening up the landscape. Human impact on the vegetation is also indicated by the increased presence of many herbs associated with human activities other than API (e.g. Apiaceae, Brassicaceae, Caryophyllaceae) (Fig 7). These plants can be found in pastures and fields browsed or cultivated for fodder [125], where animal dung is abundant. Significantly coprophilous fungi are present in this time interval, and the combined evidence suggests that the plain was devoted to agro-pastoral activities (see also [126]). The dramatic increase of Cichorieae (44%) among API at ca. 3310 BP, probably referred to an event of surface runoff during heavy rainfall, confirms a strong component of herding practices (Fig 3), which is also consistent with textual records of large-scale sheep herding to provide wool for palatially organised textile production [127]. After 3310 BP the OJC curve starts to increase due to the spread of *Olea* and, to a lesser extent, of *Juglans* confirming the expansion of tree cultivation and suggesting the onset of olive groves towards the second half of the Late Bronze Age (Fig 7). It is worth mentioning the peak of *Olea* around 3150 BP (1200 BCE). Due to the uncertainty of the chronology, the sample covers two centuries from 3254 to 3050 BP. This time frame is one hundred years before as well as after the destruction of the palaces and the breakdown of the palatial economies at ca. 1200 BCE. Given the earlier onset of *Olea* cultivation, the scale of the palatial economies and the role of olive oil on palatial documentation [128], it seems likely that the peak should be positioned during the final phase of the palatial period, indicating significant olive cultivation on the Argive Plain at the time. Nonetheless, the anthropogenic footprint for tree crops during the Mycenaean period is quite weak at Lake Lerna, both in this and Jahns' record [13], if considering the archaeological evidence from the Argive Plain itself and, more in general, from southern Greece (Fig 8). At Tiryns the study of archaeobotanical remains pointed out the well-established exploitation of olive trees [16]. In southwestern Peloponnese the pollen record of Gialova lagoon, close to the important palatial centre of Pylos, shows that the increase in *Olea* pollen starts during the Late Bronze Age and suggests the expansion of cultivated trees in the coastal plain [129]. Considering all this evidence, it is feasible that the important but weak signal of olive cultivation at Lerna during this time originates from a less widespread distribution of olive groves in the Argive Plain (cf. [2]). A similar trend in *Olea* expansion is recorded at the coastal wetland of Vravron in Attica, where the nearby settlement exponentially grew during Mycenaean times but a clear signal for cultivation started after this period [130].

From ca. 3310 to 3150 BP (1360–1200 BCE) there is an abrupt increase in the macrophytes, mainly *Typha angustifolia* type, together with NPPs growing in shallow water. $\delta D_{23}$ values reveal climatic instability with increasing humidity after a short period of dry conditions and TOC also increases (Fig 7). In the sediment succession this period is represented by gyttja with a reduced presence of silty minerals transported with other sediments by flowing water (Fig 2). The expansion of aquatic taxa reveals that the water depth in the coring site increased. The evidence for changes in sedimentation and lake productivity, in particular, seems to suggest the reduction of flow dynamic from the tributaries of the ancient lake. Large-scale modifications that could have altered the freshwater flow are known for the Argive landscape during Mycenaean times, such as the construction of agricultural terraces (to expand cultivability and secure water availability on sloping ground), the rerouting of a river near Tiryns and building

of dams at Mycenae and Tiryns [131]. The latter large projects in the eastern part of the plain show the ability of the Mycenaean societies to regulate water, and the hydrological circumstances noted in the present study could indicate that Mycenaean management of the western plain, and perhaps of Lake Lerna itself, may have been more advanced than what is currently known. Regardless of the cause, the appearance of macrophytes between the lacustrine vegetation testifies to a local palaeoenvironmental change that modified the character of the wetland until modern times.

## The end of the Late Bronze Age and the Early Iron Age (3150–2650 BP)

The Lerna $\delta D_{23}$ values indicate wetter conditions from the end of the Bronze Age and during the Early Iron Age, as also recorded in some Peloponnesian records [132–134] and in North Aegean [117]. However, other Peloponnesian and Eastern Mediterranean records show a strongly oscillating pattern with generally drier conditions interrupted by a brief wet period between 3000–2900 BP [112, 116], ca. 2800 BP [135], and ca. 2800 and 2650 BP [136], or instead a significant arid and/or cold phase throughout this period [115, 133, 137]. After a decline of macrophytes at ca. 3040–2950 BP (1090–1000 BCE), the pollen diagram profiles a pronounced peak of aquatic plants around 2880–2740 BP (930–790 BCE), primarily represented by *Nymphaea* (water lilies) (Fig 5). Nymphaeaceae are leaf-floating plants from stagnant environments typical of ponds and canals where the water column is beyond 1.5 m of height [25]. The rapid spread of these plants suggests deeper water conditions and, consequently, the growing size of the lake. This change may clearly have been driven by increased precipitation as recorded by the $\delta D_{23}$ record. Given the inconsistency between the dry trend attested by the regional climate records and the local wet conditions, other drivers cannot be excluded such as human activities changing the flow dynamics into the lake and creating permanent water conditions. As far as we know, such evidence has not been archaeologically confirmed for this period.

At ca. 3090–2620 BP (1140–670 BCE) the increase of thermophilous and montane tree pollen testifies to a new expansion of deciduous mixed and altitudinal forests (Fig 7). This reforestation could be linked to wetter conditions as well as to lower levels of human activity during the post-palatial period and the Early Iron Age [39]. The presence of *Betula* in the pollen catchment area of Lake Lerna is noteworthy (Fig 3) being already attested in the Janhs' record [13]. Its distribution is limited today to the mountains of northeastern and central Greece [138]. Nowadays, only *B. pendula* (silver birch) is present in Greece being more tolerant than the other species in terms of temperature and humidity [139, 140], even if its southern limit appears to be set by the summer drought [141]. During the Holocene and since recent times *Betula* occured sparsely in pollen diagrams from the southern Balkans (e.g. Elatia Mire, Ioannina, Lake Dojran, Lake Ohrid, Lake Prespa, Lake Skodra, Livaditis; [11, 142–147]). Its attestation in central Greece (e.g. Klisova lagoon, Lake Kopais, Lake Stymphalia; [122, 148, 149]) confirms that the region is the southeasternmost boundary of the expansion of birch in Europe. It is possible that some upland valleys of the Peloponnese could have played a role of refugia for *Betula* and possibly other temperate trees in the otherwise warm Mediterranean region [150].

## Archaic, Classical and Hellenistic period (2650–2096 BP)

A progressive decline of thermophilous taxa and the concurrent increase of Mediterranean tree pollen are recorded between ca. 2620 and 2240 BP (670–290 BCE), from the Archaic to the early Hellenistic period. Cyperaceae percentages and concentration values also decrease from ca. 2740 BP. At the same time arid conditions are suggested by the $\delta D_{23}$ record from ca.

2590 BP onwards (Fig 7). A similar climatic trend is recorded by some regional lacustrine records [134, 151] and in North Aegean [117], but other records from the Peloponnese and southwestern Anatolia suggest the prevalence of wetter conditions after ca. 2450 BP (500 BCE; [95, 112, 116, 133]). Even if climate could have influenced the vegetation development of the Argive Plain, the forest decline could also be likely explained through increasing human pressure on the landscape (see the Late Bronze Age, above), evident not least through archaeological field surveys carried out in northeastern Peloponnese [152–154]. Mediterranean plants might have dominated as the result of land degradation which caused the expansion of evergreen shrublands (see *Phillyrea*; Fig 3).

API show a general increase (Fig 7): cereals, which show a continuous curve since the Early Iron Age, reach a peak around 2450 BP corresponding to a period of population expansion in northeastern Peloponnese as well as of continuous habitation at Lerna [2, 42, 154]. The concurrent occurrence of weeds (e.g. *Polygonum aviculare* type; Fig 3) and the parasitic fungus *Ustilago* (Fig 5) confirms the presence of crop fields in the area during the Archaic period. After ca. 2450 BP, the increase of Cichorieae and *Plantago*, as well as the presence of Poaceae, suggest the prevalence of pasturelands. Such changes may reflect changing patterns of land use connected with the Lerna settlement, which continued to flourish throughout the Classical and Early Hellenistic period [42]. At the same time, Argos became a more dominant political force in the region and may have influenced a gradual change away from subsistence cultivation since the local farming seems to have been more integrated within a new regional economic framework. The slightly higher percentages of the OJC group would fit such a picture quite well, but also possibly matches a broader shift towards olive cultivation during the late Archaic and Classical period in Greece [3, 118, 119, 121, 130, 155] (for the economic background see [1]) (Figs 7 and 9). During this period, the previous palynological study [13] shows the main increase in *Olea* pollen of the entire sequence (Fig 8) and has been interpreted as an attestation of the development of arboriculture in the study area. This difference can be addressed to the different location of the coring sites. On the other hand, a critical evaluation of the radiocarbon dates from the bulk samples of Lake Lerna should be done. Although any age offset between dates on bulk and those on plant macrofossils has not been checked, the impact of a hard water effect on the clay sediments of the lake is evident from a radiocarbon age reversal shown by [34] and, more recently, from the chronological issues stressed by [29]. The organic-rich sediments instead have shown to be in proper stratigraphic order within the sequence. In the middle section of the Jahns' core an age reversal occurred as well and the inverse date from clay was excluded (see p. 191 of [13]). Considering the validity of the age-depth model of [13], we may argue that a lower sedimentation rate might have played a role in the previous core location producing a different timing for the expansion of olive cultivation in comparison with our record, whose *Olea* peak is ca. 1000 years later (see §5.5). The appearance of *Castanea* around 2360 BP (410 BCE) corresponds to the economic exploitation of such plant in Greece, traditionally dated to the Early Iron Age and increased from 500 BCE [1, 156] (Fig 3).

Since *Pinus* pollen increases from the end of the Early Iron Age onwards, we can argue that the opening up of oak woodlands and the intensification of land use attested in our record contributed to the spread of pinewoods. This progressive replacement has already been observed in previous pollen data from Lake Lerna [13] (Fig 8). Moreover, several palynological sequences from the Peloponnese (e.g. Kiladha Bay, Kleonai, Kotihi lagoon; [30, 118, 121]) and Attica (e.g. Marathon; [21]) have stressed the expansion of coastal pinewoods from ca. 3000 BP (Fig 9). The slightly higher sedimentation rate inferred from the age-depth model, the continually decreasing pollen influx from ca. 2740 BP, and the increase of erosion indicators allow us to hypothesize that the sediment infill increased in the basin (Figs 2, 4 and 5). Such a process

was probably due to continued episodes of erosion caused by the combination of potential local aridity with the expanding land use [157, 158]. The downwash of sediments into the lake culminated around 2360 BP, when the peak of Cichorieae concentration, accompanied primarily by Poaceae, attests to the transport of pollen from pastures and meadows (Fig 4). The same interpretation may be applied to the amount of degraded and indeterminable pollen grains (up to 18%). Indeed, most widespread are the incremental colluvial deposits along the foothills of the peninsula from north to south [15, 159]. Interestingly, a similar high sediment accumulation rate is recorded in this period at Lake Kournas (Crete) even if the lake eutrophication here postulated by increasing of algal remains is not attested at Lake Lerna [126]. The renewed spread of *Typha angustifolia* type in our site can hint at some changes of the water environment and it may be seen as a result of halophytic habitats, as testified by the appearance of *Tamarix* among riverine plants (Fig 3).

Still within the Hellenistic period, however, from ca. 2200 to 2120 BP (250–170 BCE) the highest AP percentages are recorded, produced by a dramatic increase in *Pinus* and *Abies*, while all the other taxa disappear with the exception of some grains of Cichorieae (Fig 7). The drop of pollen variability is confirmed by the total pollen concentration (Fig 4) and corresponds to a lithological phase of pure clay sediments, without any gyttja or silty properties as in the other layers of the Lerna core (Fig 2). The input of material from the Erasinos River would have mainly transported pollen of pine and fir from the Argive and Arcadian mountains on the west, rather than from the local environment. In addition, the pollen morphology of bisaccate grains (*Pinus* and *Abies*), as well as of Cichorieae [160], would have resulted in the selective preservation of pollen taxa. A similar but less abrupt peak of pine and fir pollen in correspondence with clay sediments is also present in the previous sequence from Lerna around 2200 BP (Fig 8). Such evidence, possibly referring to the same event in both the cores, most likely marks an erosional event: a flood might have moved sediments from degraded areas, especially on the highlands. Analogous data for pine was interpreted as a depositional effect in the mid-Holocene record of the Klisova lagoon (western Greece; [122]).

This erosional event coincides in time with a severe dry peak in the $\delta D_{23}$ record at ca. 2200 BP, after which relatively higher humidity is established although conditions generally remained drier compared to other parts of record (Fig 7). Drier conditions around 2200 BP, followed by climatic instability, are attested by some palaeoclimate proxies from the surrounding mountainous areas [133] and from North Aegean [117], while in other Peloponnesian and Eastern Mediterranean records overall wetter conditions are briefly interrupted by drier conditions around the interval 2000–1900 BP [95, 112, 116, 136]. The erosional event corresponds to the reduction in demographic pressure and to the discontinuity in the occupation of settlements reported by archaeological surveys [153]. Since palaeoclimate has been argued to be an amplifier for landscape instability primarily caused by human activity [161, 162], our evidence suggests a concurrent role of climate and field abandonment in the soil destabilisation throughout the study area.

### The Roman period (2096–1620 BP)

From the early Roman period, the decreasing Lerna $\delta D_{23}$ values testify to increasing precipitation for the period 1900–1450 BP (50–500 CE) (Fig 7). Apart from the NAO influence [163], colder and wetter conditions were generated in the northeastern Peloponnese by the NCP which led to the higher sea-effect precipitation [164, 165]. Increased moisture is recorded in many Peloponnesian records [95, 133, 151] as well as in southwestern Anatolia [116] throughout or within the 2000–1500 BP time frame. This climatic evidence coincides with a forest regeneration pattern in the Argive Plain since the highest arboreal cover is recorded in the

interval from ca. 2010 to 1660 BP (60 BCE-290 CE), with the exception of the events driven by *Pinus*. Pioneer taxa expanded preceding a new increase of deciduous oak pollen from ca. 1840 to 1660 BP accompanied by the occurrence of montane taxa (Fig 7). Such a vegetation trend is in line with high AP values observed also at other sites in the Peloponnese (e.g. Kleonai, Kotihi lagoon; [30, 121]) and in southern Greece sites (e.g. Elefsis Bay, Klisova lagoon, Vravron; [107, 122, 130]), as well as in Greece overall ca. 2000–1500 BP [3]. Since the reduction of the number of recognised pollen taxa, the low values of pollen concentration and influx and the presence of erosion indicators, the increased abundance of pollen from woodlands and highlands could have been caused by more sediment inflow from the Erasinos River due to enhanced rainfall, rather than by the growth of plant biomass (Figs 4 and 5).

The period also corresponds in time with evidence for land use changes. Although cereal pollen shows continuity, the percentages of PI and coprophilous fungi reveal that the landscape was mainly shaped by pastoral activities (Fig 7). At the same time the number of sites occurring in the Late Hellenistic to Early/Middle Roman period (ca. 150 BCE-300 CE) in the Peloponnese reduces resulting in a different spatial configuration of agricultural land use [152–155, 166]. The small settlement at Lerna also seems to have been abandoned after ca. 170 BCE [42]. It is thus possible that land use strategies were increasingly driven by landowners probably based in Argos at this time, and that this reconfiguration of land use also favoured forest regeneration.

*Olea* progressively increases and reaches 23% around 1660 BP (290 CE) (Fig 3). A similar spread of olive has been also found at Lake Kournas (Crete) in the same period as at Lerna [126]. High percentage values of *Olea* pollen have been viewed to be representative of intensive and nearby cultivation by studies on soil samples from modern olive groves [167] and moss polsters in the western Peloponnese [129]. On the contrary, an ongoing study of two new contemporary pollen sequences from Gialova lagoon suggests that *Olea* pollen is more represented in the sequence from the lake rather than in another core from the nearby peatland (Martina Hättestrand, unpublished data). For all these reasons, our data probably testifies the expansion of olive cultivation in the entire Argive Plain during the Roman period. Previously, the presence of olive pollen was attested in a decreasing trend with respect to the higher values of the Classical and Hellenistic period ([13] and Fig 8). In this respect we have already hypothesised a difference between the new and the old record from Lake Lerna due to local conditions of the coring sites (see §5.4). *Juglans*, and with a lesser extent *Cornus* and *Corylus*, also appear in our record during this time frame (Fig 3) and seem to suggest the advance of fruit-tree horticulture, as already stressed at Vravron and Elefsis Bay in Attica and at the Klisova lagoon in Messolonghi [107, 122, 130]. Palaeoenvironmental and historical studies in Greece as well as Anatolia, highlighting the Roman use of planting large-scale olive groves in coastal lowlands rather than to upland valleys, are now supported by the Lerna data [105, 107, 168]. On the other hand, a general reduction in the occurrence of *Olea* throughout southern Greece is attested during the Late Hellenistic and Roman period (e.g. [1, 3, 122, 130]), corresponding to the prevalence of cereal cultivation in general (Fig 9). As already indicated by several scholars [2, 166, 169], the Roman control on land use in the Peloponnese was based on lowland cultivation regimes, directed towards the most marketable food products (including oil, wine, and grain) mixed with extensive agro-pastoral strategies for local production and consumption [153].

## Byzantine and Medieval period (1620–487 BP)

At the beginning of the Late Roman period (also defined Early Byzantine), the decrease of $\delta D_{23}$ values continues and the lowest value is recorded at ca. 1480 BP. The lake sediment shifts

to gyttja and TOC increases suggesting lake-like conditions with enhanced productivity (Figs 2 and 7). Around 1580 BP (370 CE) a dramatic increase of Poaceae percentage and concentration together with a peak of *Typha angustifolia* type concentration are recorded and followed by peaks of concentrations of green algae and NPPs growing in shallow water (Figs 3, 4 and 6). This evidence testifies to increasing water depth, possibly due to the wetter climate conditions, corresponding to the expansion of reed vegetation. The steady increase of charcoal concentrations confirms the extensive land use both at a local and regional scale. The cultivation of olive remains important as testified by the peak in concentration and, in association to the presence in the plain of rural structures for oil production [47], reveals that the olive cultivation continued to be a large component of the landscape (Fig 4). This human pressure is also testified by archaeological remains providing evidence of the intensification of building activity at Argos in Late Antiquity [47, 48].

During the Early Byzantine period from ca. 1480 to 1120 BP (470–830 CE) the increasing percentage and influx values of *Pinus* and *Quercus robur* type evidence the expansion of both pinewoods and oakwoods in the Lerna pollen catchment area. The *Olea* curve displays a severe drop and PI significantly increases, together with *Artemisia*, Cichorieae and *Plantago* undiff. Among anthropogenic indicators (Figs 3 and 4). In addition, the archaeological evidence testifies to the shift of human occupation towards the coast, although small-scale building activity and a mixed farming and herding practice are attested throughout the entire Argolis [46, 47, 51, 170]. Therefore, pollen and archaeological data point out a reduced human pressure in the uplands and a more local food production in the plain, where olive groves contracted and pasturelands expanded following the collapse of the Eastern Roman control on the Balkans.

Apart from the decrease in olive cultivation, which was caused by the contraction of economic activities in the Argive Plain during these centuries, our record seems to suggest the relationship between reducing precipitation and increasing pastoralism in the Peloponnese as postulated by [2]. As a matter of fact, the Lerna $\delta D_{23}$ record shows a reversal of the previous trend towards wetter conditions and outlines drier conditions exactly at ca. 1450–1100 BP (500–850 CE) (Fig 7). This trend is in agreement with other lacustrine records from the Peloponnese [133, 135] and is paralleled by Eastern Mediterranean records [115, 116, 171–173], with few exceptions [168]. Notably, the speleothems used for climate reconstructions from both Mavri Trypa (ca. 1300 BP / 650 CE) and Kapsia caves (ca. 1100 BP / 850 CE) stop growing within this time frame at a time of recording very arid conditions [95, 136]. In Lake Lerna, the appearance of NPP HdV-150 [79] reveals the onset of more eutrophic conditions. Moreover, aquatic macrophytes declined and remains of green algae, as *Closterium idiosporum*, suggest the reduction of water level (Fig 5). In this respect, the Lerna evidence contributes to stress the concurrent impact of palaeoclimate on the societal development in the Mediterranean during the 1st millennium CE, as confirmed in recent reviews [172, 174].

The interval from ca. 1070 to 490 BP (880–1460 CE, the high and late Middle Ages) is characterised by a decreasing trend in *Pinus* and thermophilous trees. During this time span Poaceae pollen is abundant and does not correspond to comparable amounts of marshy taxa such as Cyperaceae (Fig 7). This evidence testifies to less forested uplands, while the plain is dominated by open environments, especially grasslands. The OJC curve reduces, whereas cereal pollen continues to be represented and its concentration values tend to increase as those of the other anthropogenic taxa of API and PI groups (e.g. Cichorieae, *Galium*, *Plantago*, *Urtica*; Fig 4). During these centuries, also called the Middle Byzantine period, the number of sites in the region increased and a picture of more intense human activity is visible through the archaeological record, even if several population deportations are attested during the 14-15th c. CE [49, 51, 53, 175]. The expansion of cereal fields and pastures in the lowlands reflect a specialized agro-pastoral economy and the intensification of activities in the rural landscape, and lead to

an opening of the Mediterranean forest [176]. Moreover, peaks of coprophilous fungal spores clearly suggest the presence of flocks in relation to animal husbandry activities (Fig 5; [177]).

During the Middle Byzantine period, the time between ca. 1100 and 800 BP (850–1150 CE) is of special interest since it corresponds to the so-called Medieval Climate Anomaly (MCA; [178–180]). The MCA is generally characterised by warmer conditions in the Balkan peninsula and parts of Anatolia but cooler conditions in the Aegean Sea and southwestern Turkey [178]. The pollen record from Lerna does not give a direct indication of warmer conditions. Evidence from the Anatolian peninsula suggests the MCA was generally wetter (e.g. [172]) but the records from Trichonida in western Greece [133] and Kocain Cave in southwestern Turkey [116] suggest the period was drier. This is similar to the local $\delta D_{23}$ record which shows drier conditions with a gradual but long-lasting return to wetter conditions after 800 BP (1150 CE) (Fig 7). Since peaks of erosion indicators are recorded after this period, we argue that the soil exposure in the previously discussed open lands could have caused slope instability and in association with increased rainfall have been responsible for soil erosion in the study area. Similar dynamics of the landscape development have been pictured out by the geomorphological and vegetation reconstruction of southern Argolis and Attica [15, 130]. In the lake, the increase of hydrophilous plants can be led by the exponential growth of *Typha latifolia* (common cattail), as recorded in the Janhs' core during a similar phase of contraction of sedge vegetation ([13] and Fig 8). Since algae and NPP remains are also present in high amounts, it is likely that precipitation caused more areas to have been inundated in the wetland and more eutrophic conditions to have been established (see HdV-150). It is worth of mention, therefore, that the common cattail is more adaptable to halophytic habitats rather than the bur-reed and narrow-leaf cattail (*Typha angustifolia* type; Fig 5).

## Ottoman and modern Greek period (487 BP-present)

From ca. 490 to 260 BP (1460–1690 CE) API and PI reach high values. As a matter of fact, pollen of cereals (up to 5%) and weeds (e.g. *Centaurea*) significantly increase in association with Cichorieae and *Plantago* (see *P. lanceolata* type), as well as Poaceae (Fig 3). The highest concentration value of microcharcoals is also recorded in this period (Fig 4). This evidence reveals that the progressive expansion of anthropogenic environments (cereal fields and pastures) and grasslands in the plain continued to shape the landscape, as witnessed in other Peloponnesian pollen diagrams (e.g. [121]). Moreover, the abundance of charcoal remains testifies to the use of fire for creating pasturelands, as well as the steady occurrence of *Sanguisorba minor* type that presumably refers to the spread of *Sarcopoterium* shrubs on the burnt lands. Such a role of human activity in shaping the landscape of the Argive Plain likely marks the economic control of the Ottoman Empire on southern Greece from ca. 550 BP (1400 CE) and the growth of population in the plain as attested by archival sources (see §2.4). As a matter of fact, the expansion of the early modern market economy in much of the Mediterranean in the 16[th] c. CE was characterised by a growing tax burden from the imperial states [181].

It is worth noting that such an increase of cereal production in the Argive Plain follows a decline in agricultural activities that pollen data from central and northern Greece has revealed for around a century after ca. 600 BP (1350 CE) due to the recurrent plague outbreaks associated with the plague bacterium *Y. pestis*, novel to this region (first introduced with the Black Death in 1346 CE) [143, 176, 182, 183]. Some authors have correlated the abandonment of crop fields in the uplands of northern Greece with a period of reduced temperature [184]. During the so-called Little Ice Age (550–250 BP/1400-1700 CE; [185]) the Lerna $\delta D_{23}$ values record wetter conditions until 360 BP and correlates with several Greek records following which it was a period of increased riverine runoff (e.g. Aegean Sea, Drama plain, Klisova

lagoon, Kotychi, Trichonida, Vravron; [122, 130, 133, 182, 186–188]; the same conditions are recorded at Kocain in southwestern Turkey, see [116]). After this, more cold and dry conditions are attested as testified by other proxies of Greece and eastern Mediterranean [115, 135, 189, 190] (Fig 7). Since the high amount of some NPPs suggest increasing eutrophic level (see HdV-150) and aquatic macrophytes start to decrease, we can postulate that the hydroclimatic changes of the LIA also contributed to reducing the size of Lake Lerna by fostering the expansion of saturated rather than submerged soils in the wetland (Fig 5).

Finally, in the period ca. 260–110 BP (1690–1840 CE) tree pollen increases by the high abundance of *Pinus*, whereas Poaceae reduces among herbs. Percentage values of PI and API remain constant, with the exception of cereals which almost disappear. Cyperaceae significantly increases among marsh vegetation and, together with the high amounts of green algae, correlates well to the short climatic trend towards wetter conditions recorded by $\delta D_{23}$ values (Fig 7). This evidence suggests the recovery of pinewoods at the expense of grasslands due to the combination of enhanced precipitation and reduced human pressure. Indeed, the 17th-18th c. CE experienced a period of political instability culminating around 1821–1832 CE with the Greek War of Independence. In this respect the presence of *Castanea* pollen among forested taxa is noticeable and likely suggests that chestnut was the most important tree cultivated during this period (Fig 3). The open landscape of the lowlands seems to have been dominated by pasturelands and/or abandoned fields, where Asteroideae and Cichorieae could also have grown as weeds. Pines could be expanded in the coastal plain as witnessed in the Vravron wetland [130], although the abrupt peak of *Pinus* pollen (and *Abies*) around 110 BP (1840 CE) might also be related to sediment inflow from the highlands due to heavy rainfall (Fig 9).

The uppermost part of the sequence (ca. 110 BP/1840 CE-present) is characterised by the exponential growth of Cichorieae pollen among API, as well as the high maxima of erosion indicators, coprophilous fungi, and NPPs growing in dry bogs (e.g. *Conidia*). Amaranthaceae pollen is also abundant among xeric plants, whereas Cyperaceae decreases together with aquatic plants and algae (Figs 3 and 5). The highest $\delta D_{23}$ values of the entire Lerna record indicate an increase in aridity leading to a reduction of the wetland area. The over-representation of Cichorieae in the last sample, representing the modern surface, may be linked to oxidation of pollen grains due to prolonged soil exposure. A similar scenario has been noted for the uppermost levels of several sediment sequences from Greece (e.g. Aliki lagoon, Elefsis Bay, Marathon, Phlious, Vravron; [21, 107, 120, 130, 159] and Fig 9). The general vegetation development of the Argive Plain was affected by the decreasing human impact, mainly represented by pastoral activities and by the continuous spread of *Pinus* and *Phillyrea*. We should also stress that the overgrazing and fires (as shown by charcoal concentration values; Fig 4) could have caused the degradation of the vegetation into garrigue, mainly composed of *Phillyrea*. Nevertheless, we can trace the exploitation and industrial expansion of the 19th c. CE to the depletion of biodiversity that still affects the ecosystems of the Argive Plain and the surrounding areas.

## Conclusions

The new pollen record from the ancient Lake Lerna provides evidence for a dynamic environmental history of the region dominated by human activities already since the mid-Holocene. Although regional palaeoclimatic fluctuations during the last 5000 years have contributed to change the ecosystems of the Argive Plain, other factors take part in the process. We prove that it was the interaction of climate with a variety of cultural, political, and socio-economic factors that transformed the vegetation and caused its long-term degradation that is testified by the progressive replacement of deciduous forests with pinewoods and open landscapes. The

environmental history of the study area was characterised by a recurring alternation of change and stability, interrupted by episodes of forest opening and ecological recovery, associated with the boom-and-bust cycles of political structures and economies of the Eastern Mediterranean. Nevertheless, we stress the role of the Peloponnese as probable refugium for temperate trees during the Holocene: the noteworthy occurrence of *Betula* in our record, which grows in the northeastern mountains of Greece nowadays, testifies to the presence of persisting ecological niches in some upland valleys despite the warm Mediterranean climate.

From the perspective of the coring site, located in the southernmost parts of the area of ancient Lake Lerna, it is possible to outline a number of phases in the history of the wetland. It is impossible to fully ascertain the size of the area covered by these changes, but our evidence shows clear hydrological and palaeoenvironmental fluctuations of potential relevance for the whole wetland. When the lowermost part of the studied sediment sequence was deposited, before ca. 4000 BP, the coring site was a fen. Thereafter the water level increased and the lake expanded over the fen, despite a regional climatic trend towards aridity. At ca. 3310 BP the abrupt expansion of macrophytes marks another significant increase of the water depth when climatic instability is recorded. The concurrent changes in sedimentation and lake productivity seem to suggest the reduction of flow dynamics and altogether may indicate that the human management of Mycenaean times produced a permanent hydrological change in Lake Lerna. What kind of modification that could have altered and increased freshwater in the lake is still unknown. During the following period varying conditions prevailed in this part of the basin and the fluctuations of water level caused the predominance of marsh over lake vegetation alternatively. Towards modern times conditions were drier and the present-day wetland was formed at the expense of the former lake.

Our results also attest to significant changes in the vegetation dynamics across the 5000 years-long study period, emphasising the effects of human activities but also the possible impact of climate variability. These developments correspond well with the vegetation history of the study area previously reconstructed by Jahns in 1993. Therefore, we argue that the two cores likely have different sedimentation rates in some levels, thus the comparison can be different in different portions of the records. In particular, the palaeoenvironmental changes recorded in the previous coring site during the Archaic-Classical-Hellenistic period are attested 1000 years later in the new sequence. In the Early Bronze Age the Argive Plain was mainly used for the cultivation of cereals by inhabitants of the nearby settlements (e.g. Lerna, Kephalari Magoula, Argos). The forest cover of the uplands was formed by mixed deciduous oakwoods, although its progressive reduction was already underway. The socio-economic growth of the Late Bronze Age palatial societies at ca. 3400–3150 BP occurred under predominantly humid climate conditions and impacted on the landscape leading to the expansion of olive groves and grazed areas throughout the plain at the expense of woodlands. After a forest recovery during the Early Iron Age, from the Archaic period (ca. 2620 BP) the expansion of open environments is attested. The increasing human pressure, potentially accentuated by local drier conditions, caused landscape instability, as attested by a dramatic alluvial event recorded in the *Pinus* curve in the Late Hellenistic period (around 2200–2120 BP / 250–170 BCE). Following the Roman conquest of Greece in 146 BCE, intensive olive cultivation expanded across the Argive Plain marking the Roman control on land use, directed towards the most marketable crop products (including olive oil). The predominant exploitation of the plain contributed to reduce the human pressure in the uplands, where oakwoods expanded again until the Middle Ages. Thereafter the progressive expansion of pinewoods both in the plain and in the uplands and the establishment of an economic landscape primarily made of pasturelands is clearly attested from ca. 1480 BP (i.e. the Byzantine period) onwards. These centuries were characterised by fluctuating palaeoclimatic conditions but changes in

vegetation correlate to a greater degree with changes in human activity rather than in climate. The degradation of vegetation in the Argive Plain is heightened by a local aridity trend in modern times, as testified in the sediment record, and contributes to cause the spread of a human-degraded environment (i.e. garrigue).

## Acknowledgments

The project is part of the Navarino Environmental Observatory (NEO) collaboration between Stockholm University, the Academy of Athens and TEMES S.S., Greece. We would like to thank the Ephorate of Antiquities of Argolida for granting the coring permit in 2016 and the Swedish Institute at Athens for help with administrative preparations. We also thank Giorgos Maneas for helping out to get permits, and Erika Modig and Helena Sunmark for assisting with the coring. Our special thanks to Karin Holmgren who initiated the project and to Laura Sadori who helped in the interpretation of pollen data.

## Author Contributions

**Conceptualization:** Cristiano Vignola, Martina Hättestrand, Adam Izdebski, Christos Katrantsiotis, Erika Weiberg, Alessia Masi.

**Formal analysis:** Cristiano Vignola, Nichola A. Strandberg.

**Funding acquisition:** Martina Hättestrand, Adam Izdebski, Elin Norström, Erika Weiberg.

**Investigation:** Cristiano Vignola, Martina Hättestrand, Christos Katrantsiotis, Nichola A. Strandberg, Alessia Masi.

**Supervision:** Martina Hättestrand, Adam Izdebski, Erika Weiberg, Alessia Masi.

**Visualization:** Cristiano Vignola, Martina Hättestrand, Christos Katrantsiotis, Erika Weiberg, Alessia Masi.

**Writing – original draft:** Cristiano Vignola, Alessia Masi.

**Writing – review & editing:** Cristiano Vignola, Martina Hättestrand, Anton Bonnier, Martin Finné, Adam Izdebski, Christos Katrantsiotis, Katerina Kouli, Georgios C. Liakopoulos, Elin Norström, Maria Papadaki, Nichola A. Strandberg, Erika Weiberg, Alessia Masi.

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
