## [Decision Letter · Decision Letter 0]

9 May 2022

PONE-D-22-09509Mid-late Holocene vegetation history of the Argive Plain (Peloponnese, Greece) as inferred from a pollen record from ancient Lake LernaPLOS ONE

Dear Dr. Vignola,

Thank you for submitting your manuscript to PLOS ONE. After careful consideration, we feel that it has merit but does not fully meet PLOS ONE’s publication criteria as it currently stands. Therefore, we invite you to submit a revised version of the manuscript that addresses the points raised during the review process.

The two reviewers greatly appreciated your work and the suggest some minor changes. Among the others, the only moderate to major revision is from Rev.1 who suggests to discuss the paper by Suzanne Jahns as a good reference for your manuscript. I agree with this suggestion and invite you to revise pat of your discussion. After that, the manuscript will be ready for acceptance.==============================

We look forward to receiving your revised manuscript.

Kind regards,

Andrea Zerboni, Ph.D.

Academic Editor

PLOS ONE

Journal Requirements:

3. We note that Figure 1 in your submission contain map/satellite images which may be copyrighted. All PLOS content is published under the Creative Commons Attribution License (CC BY 4.0), which means that the manuscript, images, and Supporting Information files will be freely available online, and any third party is permitted to access, download, copy, distribute, and use these materials in any way, even commercially, with proper attribution. For these reasons, we cannot publish previously copyrighted maps or satellite images created using proprietary data, such as Google software (Google Maps, Street View, and Earth). For more information, see our copyright guidelines: http://journals.plos.org/plosone/s/licenses-and-copyright.

a) You may seek permission from the original copyright holder of Figure 1 to publish the content specifically under the CC BY 4.0 license.  

Reviewers' comments:

Reviewer's Responses to Questions

**Comments to the Author**

1. Is the manuscript technically sound, and do the data support the conclusions?

Reviewer #1: Yes

Reviewer #2: Yes

2. Has the statistical analysis been performed appropriately and rigorously? 

Reviewer #1: N/A

Reviewer #2: N/A

3. Have the authors made all data underlying the findings in their manuscript fully available?

Reviewer #1: Yes

Reviewer #2: Yes

4. Is the manuscript presented in an intelligible fashion and written in standard English?

Reviewer #1: Yes

Reviewer #2: Yes

5. Review Comments to the Author

Reviewer #1: This is a well-executed, high quality study of landscape history in an area of notable archaeological importance, primarily based on pollen data. Two of the strengths of the study are firstly, the multi-proxy methodology which allows direct comparison with independent indicators of past climate change from the same core (as reported by Katrantsiotis et al, 2019, GloPlaCha) and, secondly, detailed evaluation of land cover change in terms of the known historical and archaeological record for this region.

Although the paper could be published as it is, I think that the authors miss an opportunity by not making systematic comparison with other pollen records from southern Greece and, in particular, with a previous pollen study from Lake Lerna by Suzanne Jahns. They state (Line 165) that “We have … chosen not to provide any direct comparisons between the old and the new record from ancient Lake Lerna”. In fact, no comparison at all is made with the Jahns study in the Discussion section of the paper! Below I outline some of the reasons why the authors should reconsider this. At the least I would like to see one or two additional diagrams comparing selected pollen taxa from the new and old cores, and potentially with other sites. This would widen reader interest in the paper and almost certainly increase its citations.

1. The old (Jahns) pollen diagram has been widely cited (>100 citations) and the data are archived in the European Pollen Database. A critical evaluation in the light of the new core would therefore aid researchers so that the old core results are not misused. Jahns’ chronology is based on 14C dates on bulk sediment samples but this does not necessarily mean that they are incorrect. In fact, as figure 2 of Vignola et al shows, there is no evidence of age offset between dates on bulk sediment and those on plant macrofossils. More important is the fact that the middle section of the old core included a 14C age reversal. Jahns selected the older 14C date as “correct” rather than the younger one, more or less arbitrarily. However, palynological comparison with the new core suggests that this may have been the wrong choice. The main rise in Olea pollen in the new core dates to the early 1st first millennium BCE, not the early 2nd millennium BCE as Jahns suggested for the old core.

2. Comparison of the two pollen cores could also help with interpretation of the new one, for example, in relation to the very abrupt change in pollen assemblages at the zone 1-2 boundary (~200 cm). Although a peak in Pinus and Abies pollen is found in both cores dating to the late 1st millennium BCE, the change is much less abrupt in the old core, suggesting that there may be a hiatus at this point in the new core. This would be supported by the δd23 curve which indicates maximum aridity at this time, potentially linked to desiccation of the ground surface at the core site.

In short, two core records may be better than one core in reconstructing regional environmental history.

In terms of the wider landscape history, the most striking mismatch with the historical-archaeology record is the surprisingly weak anthropogenic footprint during the Mycenaean period, especially for tree crops like Olea. Although the authors do allude to this, it might be worth greater emphasis. Is this also true for other pollen records near to Late Bronze Age palace settlements; e.g. Pylos?

Minor points

The authors should compare their results with another recently published study of sediment cores from Lake Lerna (Koskeridou et al 2022).

The location of other regional pollen/palaeo sites mentioned should be shown in figure 1; e.g., Stymphalia

No mention is made of δd23 methods (even if just to cite 2019 paper by Katrantsiotis in GloPlaCha)

Reference cited

Koskeridou, E., Thivaiou, D., Psarras, C., Rentoumi, E., Evelpidou, N., Saitis, G., Petropoulos, A., Ioakim, C., Katopodis, G., Papaspyropoulos, K. and Plessas, S., 2022. The Evolution of an Ancient Coastal Lake (Lerna, Peloponnese, Greece). Quaternary, 5(2), p.22.

Reviewer #2: This is an interesting paper dealing with the high-resolution study of mid-late Holocene vegetation dynamics in the Argive Plain, NE Peloponnese, aiming at evaluating the relationship between palaeoclimatic fluctuations, human impact and the evolution of the local plant ecosystem. The study area is a striking example of the long-term interaction between human communities and the environment; here, several studies on local palaeoenvironmental changes throughout the Holocene have been done (among them, the reference pollen study at Lake Lerna by Jahns, properly mentioned in the current ms that also raised the dating issues of Jahns’ record), but all lacking the details that can be provided by the current methodological advancements.

This research article presents the palynological study of the new sediment core from the ancient Lake Lerna. Based on pollen, NPP and microcharcoal analyses of 84 pollen samples covering the last 5000 years and complementary archaeological, historical and climatic data, the study establishes a cogent connection between climatic and cultural, political, and socio-economic factors in outlining the environmental history of the study area.

In this palynological paper, the Authors clearly explain the research topic and methodology; the study design is appropriate to evaluate vegetation patterns and the discussion is based on well-argued data compared to up-to-date relevant studies.

In general, beyond the suitability of the studied sediment core, the strong points of the ms are the accurate microscopic analyses and the well-argued interpretation of data that prove the good Authors’ background on past socio-environmental dynamics. The paper could represent a good reference and a new stimulating contribution to the understanding of the complexity of the Mediterranean landscape transformations.

Below, I remark just on few points/typos that should be corrected before being published:

- Chapter 2.2 = according to botanical nomenclature, please add the author’s name to the plant species name.

- Results = I would approximate the decimals of concentration values (e.g. 22,409 pollen grains/g = 22,400 p/g).

- L. 478 = ‘Ericaceae’ not in italics.

- Conclusions = I suggest including the noteworthy presence of Betula in the Lake Lerna pollen diagram.

6. PLOS authors have the option to publish the peer review history of their article (what does this mean?). If published, this will include your full peer review and any attached files.

Reviewer #1: **Yes: **Neil Roberts

Reviewer #2: No

---

## [Author Response · Author response to Decision Letter 0]

30 Jun 2022

Reviewer #1

This is a well-executed, high quality study of landscape history in an area of notable archaeological importance, primarily based on pollen data. Two of the strengths of the study are firstly, the multi-proxy methodology which allows direct comparison with independent indicators of past climate change from the same core (as reported by Katrantsiotis et al, 2019, GloPlaCha) and, secondly, detailed evaluation of land cover change in terms of the known historical and archaeological record for this region.

Although the paper could be published as it is, I think that the authors miss an opportunity by not making systematic comparison with other pollen records from southern Greece and, in particular, with a previous pollen study from Lake Lerna by Suzanne Jahns. They state (Line 165) that “We have … chosen not to provide any direct comparisons between the old and the new record from ancient Lake Lerna”. In fact, no comparison at all is made with the Jahns study in the Discussion section of the paper! Below I outline some of the reasons why the authors should reconsider this. At the least I would like to see one or two additional diagrams comparing selected pollen taxa from the new and old cores, and potentially with other sites. This would widen reader interest in the paper and almost certainly increase its citations.

1. The old (Jahns) pollen diagram has been widely cited (>100 citations) and the data are archived in the European Pollen Database. A critical evaluation in the light of the new core would therefore aid researchers so that the old core results are not misused. Jahns’ chronology is based on 14C dates on bulk sediment samples but this does not necessarily mean that they are incorrect. In fact, as figure 2 of Vignola et al shows, there is no evidence of age offset between dates on bulk sediment and those on plant macrofossils. More important is the fact that the middle section of the old core included a 14C age reversal. Jahns selected the older 14C date as “correct” rather than the younger one, more or less arbitrarily. However, palynological comparison with the new core suggests that this may have been the wrong choice. The main rise in Olea pollen in the new core dates to the early 1st first millennium BCE, not the early 2nd millennium BCE as Jahns suggested for the old core

Thank the reviewer for the comment. Actually, Jahns selected the younger date (3180 � 185 at 251-255 cm) to build her age-depth model, thus any shift between the two record is accountable. We changed the discussion of the ages from Jahns 1993 and took into account difference in local conditions of the two coring sites (e.g. sedimentation rate) starting from the useful suggestion of the reviewer. We also included comparisons with the old core throughout the text, providing a critical interpretation of the vegetation changes in the study area based on all the available data from lake Lerna. 

2. Comparison of the two pollen cores could also help with interpretation of the new one, for example, in relation to the very abrupt change in pollen assemblages at the zone 1-2 boundary (~200 cm). Although a peak in Pinus and Abies pollen is found in both cores dating to the late 1st millennium BCE, the change is much less abrupt in the old core, suggesting that there may be a hiatus at this point in the new core. This would be supported by the δd23 curve which indicates maximum aridity at this time, potentially linked to desiccation of the ground surface at the core site.

We tried to take into consideration the possibility of soil exposing in our site due to the increased aridity. But the evidence of a sediment hiatus in the new core from comparing the pine/fir peaks is not clear to us. Thanks to the reviewer’s comment, we improved the text.

In short, two core records may be better than one core in reconstructing regional environmental history.

In terms of the wider landscape history, the most striking mismatch with the historical-archaeology record is the surprisingly weak anthropogenic footprint during the Mycenaean period, especially for tree crops like Olea. Although the authors do allude to this, it might be worth greater emphasis. Is this also true for other pollen records near to Late Bronze Age palace settlements; e.g. Pylos?

The human impact during Mycenaean period in our record is mainly displayed by the reduction of thermophilous vegetation and the increase of API, suggesting the land clearance for grazing activities. We think that the scarce presence of Olea pollen at that time can be interpreted as the result of not widespread distribution of olive groves in the Argive Plain. The only exception is represented by a peak of Olea around 1200 BCE when the cultivation could have reached the southeastern part of the plain. Similar evidence has been found at Vravron in Attica, where the nearby settlement grew during Mycenaean times but the onset of olive cultivation is recorded just after this period. On the contrary, in the pollen record of Osmanaga lagoon (Gialova in the text), close to the important city-state of Pylos, the increase in Olea pollen starts during the Late Helladic and confirms the expansion of groves near the basin. We improved the text as suggested by the reviewer.

Minor points

The authors should compare their results with another recently published study of sediment cores from Lake Lerna (Koskeridou et al 2022 = Koskeridou, E., Thivaiou, D., Psarras, C., Rentoumi, E., Evelpidou, N., Saitis, G., Petropoulos, A., Ioakim, C., Katopodis, G., Papaspyropoulos, K. and Plessas, S., 2022. The Evolution of an Ancient Coastal Lake (Lerna, Peloponnese, Greece). Quaternary, 5(2), p.22.).

Thank the reviewer for presenting this new study from Lake Lerna. We referred to it in the text, although a direct comparison with our record is not possible due to 1) the lack of an age depth-model and 2) the absence of major vegetation changes in the new sequence.

The location of other regional pollen/palaeo sites mentioned should be shown in figure 1; e.g., Stymphalia

The reviewer is right, we added the other sites mentioned in Fig. 1a when they are present in the portion of Greece shown by the map.

No mention is made of δd23 methods (even if just to cite 2019 paper by Katrantsiotis in GloPlaCha)

We have referred to the original publication (Katrantsiotis et al. 2019) when mentioning the methodology for isotopic analysis in the discussion and Fig.7 caption as well.

Reviewer #2

This is an interesting paper dealing with the high-resolution study of mid-late Holocene vegetation dynamics in the Argive Plain, NE Peloponnese, aiming at evaluating the relationship between palaeoclimatic fluctuations, human impact and the evolution of the local plant ecosystem. The study area is a striking example of the long-term interaction between human communities and the environment; here, several studies on local palaeoenvironmental changes throughout the Holocene have been done (among them, the reference pollen study at Lake Lerna by Jahns, properly mentioned in the current ms that also raised the dating issues of Jahns’ record), but all lacking the details that can be provided by the current methodological advancements.

This research article presents the palynological study of the new sediment core from the ancient Lake Lerna. Based on pollen, NPP and microcharcoal analyses of 84 pollen samples covering the last 5000 years and complementary archaeological, historical and climatic data, the study establishes a cogent connection between climatic and cultural, political, and socio-economic factors in outlining the environmental history of the study area. In this palynological paper, the Authors clearly explain the research topic and methodology; the study design is appropriate to evaluate vegetation patterns and the discussion is based on well-argued data compared to up-to-date relevant studies. In general, beyond the suitability of the studied sediment core, the strong points of the ms are the accurate microscopic analyses and the well-argued interpretation of data that prove the good Authors’ background on past socio-environmental dynamics. The paper could represent a good reference and a new stimulating contribution to the understanding of the complexity of the Mediterranean landscape transformations.

Below, I remark just on few points/typos that should be corrected before being published:

Chapter 2.2 = according to botanical nomenclature, please add the author’s name to the plant species name.

Ok, done.

Results = I would approximate the decimals of concentration values (e.g. 22,409 pollen grains/g = 22,400 p/g).

Ok, done.

L. 478 = ‘Ericaceae’ not in italics.

Thanks, we modified it.

Conclusions = I suggest including the noteworthy presence of Betula in the Lake Lerna pollen diagram.

Thank the reviewer for this suggestion, we have stressed the presence of Betula in the conclusion.

---

## [Editor Report · Decision Letter 1]

4 Jul 2022

Mid-late Holocene vegetation history of the Argive Plain (Peloponnese, Greece) as inferred from a pollen record from ancient Lake Lerna

PONE-D-22-09509R1

Dear Dr. Vignola,

We’re pleased to inform you that your manuscript has been judged scientifically suitable for publication and will be formally accepted for publication once it meets all outstanding technical requirements.

Kind regards,

Andrea Zerboni, Ph.D.

Academic Editor

PLOS ONE
---

## [Editor Report · Acceptance letter]

6 Jul 2022

PONE-D-22-09509R1 

Mid-late Holocene vegetation history of the Argive Plain (Peloponnese, Greece) as inferred from a pollen record from ancient Lake Lerna 

Dear Dr. Vignola:

I'm pleased to inform you that your manuscript has been deemed suitable for publication in PLOS ONE. Congratulations! Your manuscript is now with our production department. 

Kind regards, 

on behalf of

Prof. Andrea Zerboni 

Academic Editor

PLOS ONE